# UNVEILING SCALING LAWS OF PINNS UNDER NON-EUCLIDEAN GEOMETRY

## ABSTRACT

Physics-informed neural networks (PINNs) have emerged as a powerful framework for solving partial differential equations (PDEs) by embedding physical laws into the training process. In theory, PINNs admit optimal polynomial convergence rates in approximation and generalization. However, these results rely on the unrealistic assumption of global optimization, which is intractable in practice. Consequently, scaling PINNs to large architectures remains a major challenge, as network width increases and thereby the condition number of the underlying optimization problem grows rapidly, making training increasingly difficult and creating a fundamental bottleneck. In this work, inspired by the MUON framework, we propose a descent strategy that adapts to the geometry of the optimization landscape. The new optimization algorithm does not degrade as the network size increases. As a result, we establish—for the first time—a scaling law for PINNs that predicts how performance improves systematically with model size. Using this framework, we successfully scaled PINNs to more than 1,000 neurons per layer, surpassing the conventional ranges of 200–400. This scaling perspective bridges the gap between theoretical guarantees and practical optimization, opening the door to pushing PINNs toward machine precision at unprecedented scales.

## 1 INTRODUCTION

Machine learning (ML) continues to transform science and engineering by enabling the analysis of complex data, discovery of nonlinear patterns, and development of predictive models—landmark examples include AlphaFold for protein structure prediction (Jumper et al., 2021), Deep Potentials for large-scale molecular dynamics (Zhang et al., 2018), and GraphCast for weather forecasting (Lam et al., 2023). A particularly impactful direction is physics-informed machine learning (PIML) (Karniadakis et al., 2021; Zhang et al., 2025), which integrates physical laws into ML pipelines to enhance robustness, interpretability, and generalization. Among various approaches, the most widely adopted strategy is to embed physics into loss functions. These physics-informed losses act as soft constraints, steering models to respect governing equations during training and giving rise to physics-informed neural networks (PINNs) (Raissi et al., 2019; Sirignano & Spiliopoulos, 2018; Yu et al., 2018). Owing to their flexibility and simplicity, PINNs have been applied across a wide range of scientific domains—fluid mechanics, bioengineering, material science, molecular dynamics, electromagnetics, geosciences, and thermal system design—demonstrating impressive empirical success.

Theoretical studies show that, when properly scaled and globally optimized, large PINNs can achieve optimal **scaling laws with respect to the number of collocation points**. (Duan et al., 2021; Lu et al., 2021; 2022; Ren et al., 2024) Specifically, if $N$ denotes the number of collocation points, the generalization error can decay at the optimal rate $\mathcal{E}_{\text{gen}} = O(N^{-\alpha})$, where $\alpha > 0$ depends on the smoothness of the underlying PDE solution. In practice, however, training such networks is notoriously difficult: optimization rarely approaches global optima, and issues such as spectral bias (Wang et al., 2021; Xu et al., 2019; Liu et al., 2020), gradient imbalance (Wang et al., 2022), and causality violations (Wang et al., 2024b) frequently arise. As a result, most existing works restrict attention to small, shallow networks, since larger models become severely ill-conditioned under physics-informed losses (Rathore et al., 2024). This limitation leaves the potential of deep architectures largely unexplored. This raises a natural research question:

*Can we numerically scale PINNs and observe that larger networks consistently achieve better performance, simultaneously exhibiting the predicted scaling laws?*

In this work, we show that, for large-scale PINN training, a major bottleneck lies in finding a suitable optimizer. Steepest descent with respect to a chosen norm acts as implicit preconditioning (Flynn, 2017; Maddison et al., 2021) : it rescales directions so that steps align with the landscape's geometry, accelerating convergence. In neural networks, using the spectral norm of weight matrices Carlson et al. (2015); Jordan et al. (2024) emphasizes directions of largest change in the output, effectively normalizing gradient contributions across layers and improving training stability. This approach has already been successfully applied in language models and vision tasks. (Jordan et al., 2024; Pethick et al., 2025; Liu et al., 2025)

| Method | Width | Depth |
|---|---|---|
| Vanilla PINN (Raissi et al., 2019) | 20–40 | 5–8 |
| Fourier PINNs (Wang et al., 2021) | 128–256 | 3-5 |
| FBPINNs (Moseley et al., 2023) | 16–64 | 2–5 |
| SPINN (Cho et al., 2023) | 32-256 | 3-4 |
| Causal PINNs (Wang et al., 2024b) | 128–256 | 3-5 |
| SA-PINNs (McClenny & Braga-Neto, 2023) | 50–128 | 4–6 |
| RBA-PINNs (Anagnostopoulos et al., 2023) | 128–256 | 4–6 |
| Curriculum training (Krishnapriyan et al., 2021) | 50 | 4 |
| Natural gradient descent Müller & Zeinhofer (2023); Chen et al. (2024) | 20–40 | 1-3 |
| SSBroyden (Urbán et al., 2025; Kiyani et al., 2025) | 20-40 | 2-6 |
| SOAP (Wang et al., 2025) | 256 | 6-12 |

Table 1: Representative PINN methods and typical network architectures (width = neurons per hidden layer, depth = number of hidden layers). Exact sizes may vary per problem; ranges indicate commonly reported configurations.

We propose a scaled version of MUON that, for the first time, reveals a clear scaling law for PINNs: wider neural networks consistently achieve better performance, and the improvement follows a predictable trend. Moreover, our analysis establishes a relationship between computational cost and network width, enabling us to identify compute-optimal architectures. Using this approach, we successfully scaled a PINN to more than 1,000 neurons per layer, surpassing the previous conventional limits of only 200–400 neurons.

## 1.1 RELATED WORKS

**Optimization for Physic-Informed Neural Network** (Urbán et al., 2025; Kiyani et al., 2025) employ quasi-Newton iteration algorithms, which can be interpreted within the framework of self-scaled Broyden methods. However, such quasi-Newton approaches are sensitive to noise (Xie et al., 2020; Shi et al., 2022), necessitating full-batch training. Additionally, the Broyden-family methods explicitly maintain a dense curvature matrix, resulting in a storage complexity that is quadratic in the number of parameters. Consequently, (Urbán et al., 2025; Kiyani et al., 2025) are limited to training PINNs with 8 hidden layers of 20 neurons each. Another line of research leverages natural gradient descent for training PINNs (Müller & Zeinhofer, 2023; Chen et al., 2024); these methods require computing an expansive Hessian inverse, incurring cubic computational cost per iteration with respect to the number of parameters, which restricts them to networks with 3–7 hidden layers of width 50. Other training methods for PINN include the multistage Adam+L-BFGS methods (Wang & Lai, 2024; Rathore et al., 2024) or the SOAP optimizer (Wang et al., 2021). However, their numerical performance remains limited to hidden layers containing only 200–400 neurons. Other works have demonstrated that methods such as KFAC can train substantially larger networks (Dangel et al., 2024), but require an implementation that is tightly coupled to the underlying PDE.

**Optimization for Modern Machine Learning** Modern large-scale neural networks and language models commonly rely on approximations of the AdaGrad (Duchi et al., 2011; Ward et al., 2020) algorithm. These approximations often involve diagonal preconditioning, as in Adam (Kingma & Ba, 2014; Reddi et al., 2019) and Adam-Mini (Zhang et al., 2024), or more structured tensor-based

approaches, such as Shampoo (Gupta et al., 2018) and SOAP (Vyas et al., 2024). Adam is known to perform well with low-precision computations for PINN training (Wang & Lai, 2024; Rathore et al., 2024), while recent work has employed SOAP to enable the scaling of PINN training to larger models (Wang et al., 2025). In this paper, we focus on an alternative family of solvers that perform steepest descent with respect to different norms (Flynn, 2017; Maddison et al., 2021). This includes using the $\ell_\infty$ norm, which gives rise to SignSGD (Bernstein et al., 2018), and using the spectral norm, which leads to Muon (Carlson et al., 2015; Tuddenham et al., 2022; Bernstein & Newhouse, 2024; Jordan et al., 2024; Pethick et al., 2025; Liu et al., 2025).

**Neural Scaling Law**   Recent empirical studies across speech, vision, and text modalities have shown that neural network performance follows a power-law scaling with respect to both model size and dataset size (Cortes et al., 1993; Hestness et al., 2017; Kaplan et al., 2020; Alabdulmohsin et al., 2022; Hoffmann et al., 2022; Caballero et al., 2022). This behavior can be captured by the decomposition of the generalization loss into an irreducible error term and scaling terms that decay as power laws in the network size and dataset size: $\mathcal{L} = \text{irreducible error} + \frac{A}{(\text{network size})^\alpha} + \frac{B}{(\text{data size})^\beta}$, where the exponents $\alpha$ and $\beta$ depend on the intrinsic dimension of the data manifold (Bahri et al., 2024). Theoretical analyses suggest that the scaling exponents decrease with increasing intrinsic data dimension, reflecting the slower reduction in generalization error for more complex data. Consequently, the generalization error exhibits a predictable power-law dependence on both training dataset size and network size.

## 1.2 Contribution

- We show that, unlike previous optimizers where larger PINNs fail to train effectively, using the Muon-family optimizer allows us to numerically scale PINNs for the first time and observe that larger networks consistently achieve better performance, revealing the predicted scaling laws that describe how performance systematically improves with increasing network width or capacity.

- In our work, by using the Muon optimizer to enable proper scaling, we successfully trained a PINN with a width exceeding 1000, whereas previous state-of-the-art networks could only be trained up to widths of 200–400 (Rathore et al., 2024; Wang et al., 2025), showing that only with appropriate scaling can larger PINNs be effectively optimized.

- We show that our rescaled operator-norm–constrained neural networks remain universal approximators for Sobolev functions. Importantly, the norm constraint is **independent of network width**, whereas in the Euclidean (Frobenius) geometry, the approximation norm grows with width. This demonstrates that **our rescaled operator norm provides a more meaningful description of neural networks for approximating Sobolev functions.** We also theoretically show that MUON addresses this by carefully choosing appropriate norms for each layer, which effectively controls the size of the updates. Specifically, the magnitude of updates in MUON corresponds to the nuclear norm of the gradient, and for PINNs, this nuclear norm can be effectively bounded.

**Notations.**   For $h \in \mathbb{R}^m$, we define the RMS (root-mean-square) norm as $\|h\|_{\mathrm{RMS}} := \sqrt{\frac{1}{m}\|h\|_2^2} = \frac{\|h\|_2}{\sqrt{m}}$. For a linear operator $U : A \to B$, the operator norm is defined by $\|U\|_{\mathrm{op},A \to B} := \sup_{\substack{x \in A \\ x \neq 0}} \frac{\|Ux\|_B}{\|x\|_A}$. Accordingly, we introduce the following three operator norms, which will be used in our optimizers:

1. The operator norm from $\ell_1$ to RMS: $\|W_1\|_{\mathrm{op},1 \to \mathrm{R}} := \sup_{\|x\|_1 = 1} \|W_1 x\|_{\mathrm{RMS}} = \frac{\|W_1\|}{\sqrt{m}}$.
2. The usual spectral norm, which can be equivalently written as $\|W_2\|_{\mathrm{op},\|\cdot\|_{\mathrm{RMS}} \to \|\cdot\|_{\mathrm{RMS}}}$.
3. The operator norm from RMS to $\ell^\infty$ : $\|W_3\|_{\mathrm{op},\mathrm{R} \to \infty} := \sup_{\|h\|_{\mathrm{RMS}} = 1} \|W_3 h\|_\infty = \sqrt{m}\,\|W_3\|$.

## 2 Overview of Physics-informed Neural Networks

We briefly review the standard formulation of physics-informed neural networks (PINNs) (Raissi et al., 2019). Consider a general PDE of the form

$$\mathbf{u}_t + \mathcal{D}[\mathbf{u}] = \mathbf{f}, \tag{1}$$

defined on a spatial-temporal domain $[0, T] \times \Omega \subset \mathbb{R}^{1+d}$, where $\Omega$ is a bounded domain in $\mathbb{R}^d$ with regular enough boundary $\partial\Omega$, $\mathcal{D}[\cdot]$ is a linear or nonlinear differential operator, and $\mathbf{u}(t, \mathbf{x})$ denotes a unknown solution. The initial and boundary conditions are

$$\mathbf{u}(0, \mathbf{x}) = \mathbf{g}(\mathbf{x}), \quad \mathbf{x} \in \Omega, \tag{2}$$

$$\mathcal{B}[\mathbf{u}] = 0, \quad (t, \mathbf{x}) \in [0, T] \times \partial\Omega, \tag{3}$$

where $\mathbf{f}$ and $\mathbf{g}$ are given functions, and $\mathcal{B}[\cdot]$ denotes a general boundary operator (e.g., Dirichlet, Neumann, Robin, or periodic). Under standard assumptions, problem equation 1–equation 3 is well-posed.

PINNs approximate the solution by a neural network $\mathbf{u}_\theta(t, \mathbf{x})$ with parameters $\theta$. The network is trained by minimizing a composite loss function:

$$\mathcal{L}(\theta) = \underbrace{\int_\Omega |\mathcal{I}[\mathbf{u}_\theta](\mathbf{x})|^2 \, d\mathbf{x}}_{\mathcal{L}_{\text{ic}}(\theta)} + \underbrace{\int_0^T \int_{\partial\Omega} |\mathcal{B}[\mathbf{u}_\theta](t, \mathbf{x})|^2 \, d\mathbf{x} \, dt}_{\mathcal{L}_{\text{bc}}(\theta)} + \underbrace{\int_0^T \int_\Omega |\mathcal{R}[\mathbf{u}_\theta](t, \mathbf{x})|^2 \, d\mathbf{x} \, dt}_{\mathcal{L}_{\text{pde}}(\theta)},$$
$$\tag{4}$$

where $\mathcal{I}[\cdot]$, $\mathcal{B}[\cdot]$, and $\mathcal{R}[\cdot]$ denote the operators for initial conditions, boundary conditions, and PDE residuals, respectively.

In practice, the integrals are approximated via Monte Carlo sampling at collocation points: $\{\mathbf{x}_i^{\text{ic}}\}_{i=1}^{N_{\text{ic}}} \subset \Omega$ for $\mathcal{L}_{\text{ic}}$, $\{(t_i^{\text{bc}}, \mathbf{x}_i^{\text{bc}})\}_{i=1}^{N_{\text{bc}}} \subset [0, T] \times \partial\Omega$ for $\mathcal{L}_{\text{bc}}$, and $\{(t_i^{\text{pde}}, \mathbf{x}_i^{\text{pde}})\}_{i=1}^{N_{\text{pde}}} \subset [0, T] \times \Omega$ for $\mathcal{L}_{\text{pde}}$. The total loss is the empirical average over these samples, and network parameters $\theta$ are optimized via stochastic gradient descent.

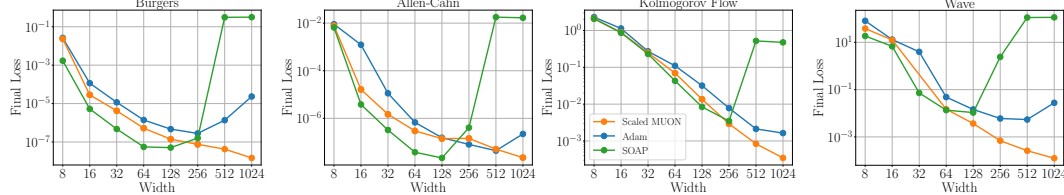

Figure 1: **Larger PINN is Harder to Train.** We compare how different optimizers perform for different network sizes. We plot the final loss achieved after training against the network width. Larger PINN is harder to train. This contrasts with deep learning, where wider networks are often easier to train.

## 2.1 SCALABILITY ISSUE

Training PINNs has proven to be more challenging than conventional neural networks, with several key issues identified in recent research. Training PINNs is extremely challenging because it requires highly accurate optimization, scale-independent treatment of different loss terms (Wang et al., 2022), and the underlying physical equations often induce extremely large condition numbers (Rathore et al., 2024). Due to these challenges, the practical use of PINNs has mostly been restricted to relatively small networks—**typically no more than 5 layers and a width of 200-400** (Rathore et al., 2024; Wang et al., 2025). This stands in sharp contrast to the remarkable phenomenon in the broader field of deep learning, where wider networks become easier to train. In fact, for PINNs, increasing the network size often leads to worse accuracy and stability, rather than improvements (Wang et al., 2024a). As a result, **PINNs fail to exhibit the predictable scaling laws commonly observed in modern deep learning**, where larger models consistently yield better performance (Hestness et al., 2017; Kaplan et al., 2020).

From a theoretical perspective, prior work has shown that, when properly scaled and globally optimized, large PINNs can achieve optimal scaling laws with respect to the number of collocation points (Duan et al., 2021; Lu et al., 2021; 2022; Ren et al., 2024). Specifically, if $N$ denotes the number of collocation points, the generalization error can decay at the optimal rate $\mathcal{E}_{\text{gen}} = O(N^{-\alpha})$, where $\alpha > 0$ depends on the smoothness of the underlying PDE solution. Since the overall risk

of a neural network can typically be decomposed into approximation, generalization, and optimization errors, and both the approximation and generalization components of PINNs are theoretically well behaved, we hypothesize that the primary bottleneck in practical applications arises from the optimization error.

In the following, we empirically and theoretically show that the trainability of PINNs deteriorates as the network width increases, especially when the state-of-the-art Piratenets Wang et al. (2024a) are used as the backbone architecture. This behavior is consistent with observations that the optimization landscape of PINNs is particularly ill-conditioned (Rathore et al., 2024). Figure 1 shows the resulting relative PINNs when training PINNs with MLPs of varying width. The prediction error of previous optimizers such as Adam and SOAP increases with network width.

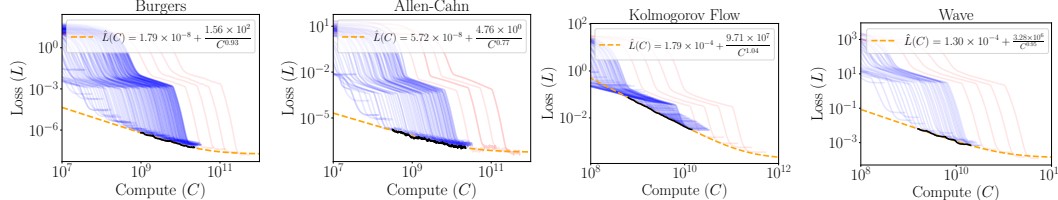

Figure 2: **Scaling laws for Burgers, Allen–Cahn, Kolmogorov Flow, and Wave equation.** Using the scaled MUON optimizer, we train networks on widths ranging from 8 to 1024. From loss curves with width less than 256 (blue), we calculate the minimal loss at each level of compute (black). We use this compute-optimal loss curve to fit a scaling law (orange), validating the fit using loss curves with width greater than 256 (red).

## 3 METHODOLOGY

Gradient-based optimization constitutes a fundamental approach for minimizing differentiable functions. Classical steepest descent updates parameters along the negative gradient of the objective, which corresponds to the direction of maximal instantaneous decrease measured in the Euclidean norm: $x_{k+1} = x_k - \eta_k \nabla f(x_k)$, where $\eta_k > 0$ denotes the step size. The gradient $\nabla f(x_k)$ is the steepest descent direction under the $\ell_2$ norm because, among all unit-length directions in this norm, it maximizes the rate of decrease of $f$: $\nabla f(x_k) = \arg\max_{\|d\|_2=1} \nabla f(x_k)^\top d$, i.e., it is the unit-norm direction along which the directional derivative of $f$ is maximized.

More generally, steepest descent can be defined with respect to an arbitrary norm $\|\cdot\|$. At iteration $k$, the steepest descent direction $d_k$ is the unit-norm direction achieving the maximal instantaneous decrease in the objective: $d_k := \arg\min_{\|d\|=1} \nabla f(x_k)^\top d$. The steepest descent direction can be expressed as $d_k \in \partial \|\nabla f(x_k)\|_*$, where $\partial\|\cdot\|_*$ denotes the subdifferential of the dual norm. (Flynn, 2017; Maddison et al., 2021) This formulation recovers classical Euclidean steepest descent when $\|\cdot\| = \|\cdot\|_2$, while alternative norms induce different update rules that exploit problem-specific geometry. For instance, the $\ell_\infty$ norm leads to sign-based updates (Bernstein et al., 2018), and operator norm choices yield whitened descent directions (Jordan et al., 2024). Adopting steepest descent under alternative norms provides a principled mechanism to mitigate ill-conditioning and improve convergence behavior in structured optimization problems.

In scaled MUON, we apply steepest descent updates under different norms depending on the parameter type. For the first-layer weight matrices, we use the operator norm $\|\cdot\|_{\mathrm{op},1\to\mathrm{RMS}}$. For hidden-layer weight matrices, we use the spectral norm $\|\cdot\|_{\mathrm{op},\mathrm{RMS}\to\mathrm{RMS}}$. For the last-layer weight matrix, we adopt the operator norm $\|\cdot\|_{\mathrm{op},\mathrm{RMS}\to\infty}$. Bias vectors are updated using the RMS norm. This design naturally scales up the learning rate in the first layer and scales it down in the last layer as the network width increases. Unlike the conventional MUON scheme, which combines spectral updates with Adam for the last-layer and bias parameters, scaled MUON applies tailored norm-based updates consistently across all parameter types. Algorithm 1 illustrates the Scaled MUON optimizer. We show how these norm yield the

**Difference with MUON.** Our scaled MUON differs from the original MUON in two key aspects. First, the original MUON applies steepest descent under the spectral norm to the matrix layers while

using Adam for the first and last layers. In contrast, we apply steepest descent under a designed norm across all layers. Second, whereas MUON uses the $\ell_2$ norm, we adopt the RMS norm for hidden features, which introduces a width-dependent scaling of the updates. This makes the last-layer updates smaller and the first-layer updates larger. As shown in Figure 4, this scaling of update magnitudes allows us to satisfy the spectral condition of Yang et al. (2023), thereby ensuring stable feature learning. As shown in Figure 3, Scaled MUON scales PINNs, whereas MUON does not. In section 5, we further demonstrate that the rescaled norm provides an effective framework for describing function approximation.

---

**Algorithm 1** Scaled MUON (The blue comments highlights the difference with MUON)

---

**Require:** Learning rate sequence $\{\eta^{(t)}\}$, momentum parameter $\alpha$, initial parameter $\Theta^{(0)}$

    $M^{(0)} \leftarrow \mathbf{0}$
    **for** $t = 1, 2, \ldots$ **do**
        $G \leftarrow \nabla L(\Theta^{(t-1)})$
        $M^{(t)} \leftarrow \alpha M^{(t-1)} + (1 - \alpha)G$
        $\tilde{G} \leftarrow (1 - \alpha)G + \alpha M^{(t)}$
        **if** $\Theta^{(t-1)} \in \mathbb{R}^{m \times n}$ is weight matrix in first layer **then**
            **for** $j = 1$ to $n$ **do**
                $\Theta^{(t)}_{:,j} \leftarrow \Theta^{(t-1)}_{:,j} - \eta^{(t)} \, m^{1/2} \, \tilde{G}_{:,j} \, / \, \|\tilde{G}_{:,j}\|_2$         ▷ Scales up the learning rate.
        **else if** $\Theta^{(t-1)} \in \mathbb{R}^{m \times n}$ is weight matrix in hidden layer **then**
            $\Theta^{(t)} \leftarrow \Theta^{(t-1)} - \eta^{(t)} \, \mathrm{signm}(\tilde{G})$         ▷ $\mathrm{signm}(A) = A(A^\top A)^{-1/2}$
        **else if** $\Theta^{(t-1)} \in \mathbb{R}^{m \times n}$ is weight matrix in last layer **then**
            **for** $i = 1$ to $m$ **do**
                $\Theta^{(t)}_{i,:} \leftarrow \Theta^{(t-1)}_{i,:} - \eta^{(t)} \, n^{-1/2} \, \tilde{G}_{i,:} \, / \, \|\tilde{G}_{i,:}\|_2$         ▷ Scales down the learning rate.
        **else if** $\Theta^{(t-1)} \in \mathbb{R}^n$ is bias vector **then**
            $\Theta^{(t)} \leftarrow \Theta^{(t-1)} - \eta^{(t)} \, \tilde{G} \, / \, \|\tilde{G}\|_{\mathrm{RMS}}$
        **else if** $\Theta^{(t-1)} \in \mathbb{R}$ is scalar parameter **then**
            $\Theta^{(t)} \leftarrow \Theta^{(t-1)} - \eta^{(t)} \, \mathrm{sign}(\tilde{G})$

---

## 4 EXPERIMENTS

In our experiments, we demonstrate that Scaled MUON's ability to train PINNs at large widths enables the construction of scaling laws. These scaling laws reveal that Scaled MUON not only makes wide-network training feasible but also ensures consistent loss reduction as width grows. To illustrate this, we construct scaling laws for four benchmark problems: Burgers equation, Allen-Cahn equation, Kolmogorov flow, and the Wave equation. Additionally, we evaluate Scaled MUON on a high-dimensional Poisson equation to assess its computational efficiency.

**Wave equation.** The one-dimensional wave equation is given by

$$u_{tt} = c^2 u_{xx},$$

where $u(x, t)$ denotes the wave field and $c = 2$ is the wave speed. We consider the solution on the domain $(x, t) \in \Omega = [0, 1] \times [0, 1]$, subject to the initial conditions $u(x, 0) = \sin(\pi x) + \frac{1}{2} \sin(5\pi x)$ and $u_t(x, 0) = 0$ for $x \in [-1, 1]$, and Dirichlet boundary conditions $u(0, t) = u(1, t) = 0$ for $t \in [0, 1]$.

**Burgers equation.** The viscous Burgers equation is given by

$$u_t + u u_x = \nu u_{xx},$$

where $u$ is the velocity field, and $\nu$ is the viscosity coefficient, which controls the strength of the diffusive term $u_{xx}$. In our experiments, we consider the domain $(x, t) \in \Omega = [-1, 1] \times [0, 1]$, with initial condition $u(x, 0) = -\sin(\pi x)$ for $x \in [-1, 1]$, and Dirichlet boundary conditions $u(-1, t) = u(1, t) = 0$ for $t \in [0, 1]$. We set $\nu = 0.01/\pi$.

**Allen-Cahn equation.** The Allen-Cahn equation is given by

$$u_t = \epsilon^2 u_{xx} + f(u),$$

where $f(u)$ is the reaction term, and $\epsilon$ is the interface thickness parameter. In our experiments, we set $f(u) = 5(u - u^3)$ and $\epsilon = 0.01$. We consider the solution on the domain $(x, t) \in \Omega = [-1, 1] \times [0, 1]$, subject to initial conditions $u(x, 0) = x^2 \cos(\pi x)$ for $x \in [-1, 1]$, and periodic boundary conditions $u(-1, t) = u(1, t)$ and $u_x(-1, t) = u_x(1, t)$ for $t \in [0, 1]$.

**Kolmogorov flow.** The two-dimensional Kolmogorov flow is governed by the Navier-Stokes equations

$$\partial_t \mathbf{u} + (\mathbf{u} \cdot \nabla)\mathbf{u} = -\nabla p + \nu \Delta \mathbf{u} + \mathbf{f}, \qquad \nabla \cdot \mathbf{u} = 0,$$

where $\mathbf{u}(x, y, t) = (u, v)$ is the velocity field, $p(x, y, t)$ is the pressure, and $\nu = 0.0005$ is the viscosity. The forcing is $\mathbf{f}(x, y) = \big(0, \, 2\sin(4\pi y)\big)$, and the solution is considered on the periodic domain $(x, y) \in [0, 1] \times [0, 1]$. We impose a random initial velocity field at $t = 0$ and study the evolution of the system over the time interval $t \in [0, 0.1]$.

**100D Poisson equation.** On the domain $\Omega = [0, 1]^{100}$, we define the exact solution as $u(\mathbf{x}) = \|\mathbf{x}\|^2$. Then the Poisson equation is given by $-\Delta u(\mathbf{x}) = -200, \quad \mathbf{x} \in \Omega$, with Dirichlet boundary conditions $u(\mathbf{x}) = \|\mathbf{x}\|^2$ for $\mathbf{x} \in \partial\Omega$.

**Baselines.** In all experiments, we primarily adopt PirateNet (Wang et al., 2024a) as the backbone architecture, trained with either the default Adam optimizer or SOAP (Vyas et al., 2024). PirateNet has been shown to scale effectively with network depth, while SOAP achieves faster and more stable convergence in PINN training (Wang et al., 2025).

**Training setup.** For all experiments, we use the PirateNet architecture with a random Fourier feature layer, which contains no trainable parameters and reduces bias toward low-frequency solutions. Each network is trained for 100,000 steps. At every step, we sample 2,048 collocation points from the interior of the domain, 512 points from the initial condition. If the benchmark problem has periodic boundary conditions with period $P$, we transform the input from $(x, t)$ to $(\mathbf{v}(x), t)$, where $\mathbf{v}(x) = [\cos(2\pi x/P), \sin(2\pi x/P)]$, ensuring that the boundary conditions are satisfied explicitly. The learning rate schedule consists of an initial warmup phase of 20,000 steps at a constant rate of $10^{-3}$, followed by 60,000 steps of exponential decay down to $10^{-8}$, and a final 20,000 steps at a constant rate of $10^{-8}$.

### 4.1 CONSTRUCTING THE SCALING LAW

**Scaling law.** With a method of estimating compute, we now estimate the compute-optimal loss $L_{\min}(C)$, the minimal loss able to be achieved at a given compute $C$. To model the compute-optimal loss, we consider a scaling law of the form

$$\hat{L}_{\min}(C) = L_0 + \frac{A}{C^\alpha}, \tag{5}$$

where parameter $L_0$ represents the irreducible error, and parameters $A$ and $\alpha$ control the shape of the function.

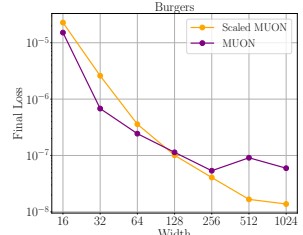

Figure 3: Scaled MUON shows more predictable trend in final loss across widths than in MUON.

The resulting scaling laws and process to estimate the parameters in equation 5 are summarized in figure 2. To generate datapoints to learn the parameters in equation 5, we use scaled MUON to optimize networks of widths ranging from 8 to 256, recording the loss curves. Using an estimate of compute and interpolating, we get the loss of each network as a function of compute. After applying Gaussian smoothing, we calculate the compute-optimal loss on an interval of compute values, and we use these data points to estimate $L_0$, $A$, and $\alpha$ in equation 5. Additionally, we train networks of widths ranging from 256 to 1024 to validate that the learned scaling law extrapolates to larger compute. We show in Figure 2 that the scaling law predicts large-scale training behavior.

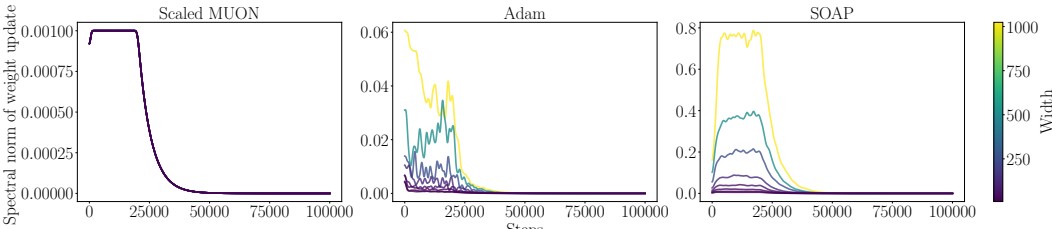

Figure 4: **Spectral norms of weight updates.** We show that only Scaled MUON satisfies the spectral condition in (Yang et al., 2023), whereas the update scales of other optimizers diverge as width increases, leading to unstable training behavior. All training runs follow the same learning rate schedule, with 20,000 steps of warmup and 80,000 steps of exponential decay.

### 4.2 COST OF SCALED MUON

**Computational complexity of Scaled MUON** We show that the computational complexity of Scaled MUON can scale to large networks. According to algorithm 1, different parameter groups follow different update rules. Updating the first and last weight layers has cost $O(mn)$, while updating bias parameters has cost $O(n)$, since these updates require only vector norm computations. The dominant cost of Scaled MUON arises from computing the matrix sign function $\mathrm{signm}(\tilde{G})$ for hidden layer updates. We approximate $\mathrm{signm}(\tilde{G})$ using the Newton-Schulz iteration in algorithm 2. Each Newton-Schulz step involves matrix-matrix multiplications, so for square hidden layers the total cost of evaluating $\mathrm{signm}(\tilde{G})$ is $O(Km^3)$, where $K$ denotes the number of Newton-Schulz iterations. Although cubic in width, these operations are highly optimized on GPUs. For memory complexity, Scaled MUON maintains a single first-order momentum buffer and does not maintain any second-order statistics, resulting in a lightweight memory usage suitable for large networks.

**Comparing Scaled MUON with KFAC on 100D Poisson.** To compare Scaled MUON with KFAC, we run them on the 100D Poisson example on different widths at a fixed time budget of 3600 seconds. Table 2 shows that a single step of Scaled MUON is computationally cheaper than a single step of KFAC, resulting in more optimization steps in the same time span. Figure 5 further demonstrates that Scaled MUON reaches a given relative error more quickly than KFAC.

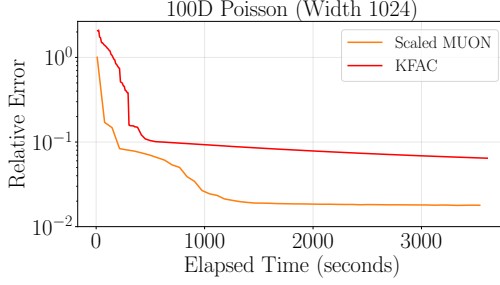

| Width | Scaled MUON | | KFAC | |
|---|---|---|---|---|
| | Steps | Rel. Err. | Steps | Rel. Err. |
| 8 | 202k | 0.045 | 55.1k | 0.195 |
| 16 | 163k | 0.035 | 44.1k | 0.132 |
| 32 | 100k | 0.028 | 24.0k | 0.085 |
| 64 | 59.2k | 0.019 | 11.1k | 0.073 |
| 128 | 33.9k | 0.008 | 5.60k | 0.073 |
| 256 | 17.6k | 0.011 | 2.67k | 0.057 |
| 512 | 7.45k | 0.013 | 1.11k | 0.059 |
| 1024 | 2.59k | 0.018 | 0.41k | 0.064 |

Figure 5: Scaled MUON attains lower relative error compared to KFAC over time at width 1024 on the 100D Poisson problem with a fixed time budget of 3600 seconds.

Table 2: Comparison of final relative error and number of optimizer steps on 100D Poisson problem. In a fixed time span of 3600 seconds, Scaled MUON performs 4-7x more steps than KFAC.

## 5 WHY SPECTRAL NORM IS A BETTER NORM?

In this section, we aim to explain why spectral norm is a good norm for PINN. Chen et al. (2025) shows that MUON implicitly solves an optimization problem that enforces a constraint on the spectral norm of the weight matrices. Naturally, we would raise the following questions

- Does a neural network remain a universal approximator when its spectral norm is constrained?
- Why is a spectral norm constrained PINN easier to optimize?

In this section, we demonstrate that a neural network can still approximate all functions in the Sobolev space even when its spectral norm is bounded. Moreover, we show that under the scaled norm constraint, we can produce more stable updates.

Define the neural network with depth $K$, width $W$, sparsity constraint $S$, and norm constraint $B$ as the class of functions

$$
\mathcal{F}_\sigma(K, m, B) := \left\{ f(x; \boldsymbol{W}_{1:K}, b) = y^K(x) \, \middle| \, \begin{array}{l} y^0(x) = x, \\ y^i(x) = \sigma(\boldsymbol{W}_i y^{i-1}(x) + b_i), \ 1 \le i < L, \\ y^K(x) = \boldsymbol{W}_K y^{K-1}(x) + b_K, \\ \boldsymbol{W}^{(1)} \in \mathbb{R}^{m \times d}, \ b^{(1)} \in \mathbb{R}^m, \\ \boldsymbol{W}^{(i)} \in \mathbb{R}^{m \times m}, \ b^{(i)} \in \mathbb{R}^m, \ 1 < i < L, \\ \boldsymbol{W}^{(L)} \in \mathbb{R}^{1 \times m}, \ b^{(L)} \in \mathbb{R}, \\ \|\boldsymbol{W}_{1:K}\|_{\text{block}} \le B \end{array} \right\},
$$

where the block norm is defined as $\|\boldsymbol{W}_{1:K}\|_{\text{block}} := \|\boldsymbol{W}_1\|_{\text{op},1\to\text{RMS}} + \sum_{i=2}^{K-1} \|\boldsymbol{W}_i\|_{\text{op},\text{RMS}\to\text{RMS}} + \|\boldsymbol{W}_K\|_{\text{op},\text{RMS}\to\infty}$. We then show that neural networks with spectral norm constraints can still approximate all functions in the Sobolev space

**Theorem 1** (Norm Constrained Universal Approximation). *Fix dimension $d \in \mathbb{Z}^+$ and domain $\Omega \subseteq [0,1]^d$, for any $s \in \mathbb{Z}^+$ and any function $u^* \in H^s(\Omega)$, there exists a spectral norm constrained Deep Neural Network $u_{DNN} \in \mathcal{F}_{ReLU^3}(K, m, B)$ with $K = O(1), m = O(N), B = O(1)$, such that:*

$$
\|u_{DNN} - u^*\|_{L^2(\Omega)} \lesssim N^{-\frac{s}{d}} \|u^*\|_{H^s(\Omega)}. \tag{6}
$$

**Remark 1.** *Previous approximation theory (Yarotsky, 2017; Suzuki, 2018; Lu et al., 2021) considered neural network weights under the vector $\ell_\infty$ geometry, showing that the weight bound scales as $O(m)$, and the Frobenius norm of the constructed networks also scales as $O(m)$. These results suggest that using **our scaled operator norm is also effective for neural networks approximating Sobolev functions**.*

The theorem demonstrates that even when the spectral norm is fixed, the neural network can still approximate all functions in the Sobolev space; that is, the spectral norm required to approximate Sobolev functions does not need to increase with the network width. Consequently, **optimization under a spectral norm constraint can still recover the underlying ground-truth functions**.

The next question the paper aims to address is why optimization under a spectral norm constraint is more suitable for PINNs. To demonstrate that, we prove that the nuclear norm, the dual norm of operator norm, of the PINN gradient can be bounded.

**Why Nuclear Norm of Gradient is Important?** When performing steepest descent to optimize function $f$ with respect to the operator norm, the update decreases the objective $f(X)$:

$$
\Delta f = -\langle \nabla f(X), \text{signm}(\nabla f(X)) \rangle = -\|\nabla f(X)\|_*,
$$

where $\|\cdot\|_*$ denotes the nuclear norm. Therefore, the decrease in the objective induced by an operator-norm steepest descent step is exactly the nuclear norm of the gradient. **This shows that if the nuclear norm of the gradient is unbounded, the update to the objective function can become extremely unstable.** Next, we demonstrate that the nuclear norm of the gradient of the PINN objective can be bounded.

**Theorem 2** (Scale of PINN Gradient). *Let $y^i(\boldsymbol{W}_{1:K}; x)$ be a feedforward neural network defined recursively as*

$$
y^i(\boldsymbol{W}_{1:K}; x) = \begin{cases} h(y^{i-1}(\boldsymbol{W}_{1:i-1}; x), \boldsymbol{W}_i), & 2 \le i \le K, \\ h(x, \boldsymbol{W}_1), & i = 1, \end{cases} \tag{7}
$$

*where $x \in \mathbb{R}^d$ is the input, $y^K(\boldsymbol{W}_{1:K}; x) \in \mathbb{R}$ is the output, and $\boldsymbol{W}_{1:K} = (\boldsymbol{W}_1, \ldots, \boldsymbol{W}_K)$ collects all network parameters, here $\boldsymbol{W}_1 \in \mathbb{R}^{m \times d}, \boldsymbol{W}_i \in \mathbb{R}^{m \times m}, \boldsymbol{W}_K \in \mathbb{R}^{1 \times m} (2 \leq i \leq K - 1)$ where $m$ is the network width.*

*Consider a PINN objective function $f(\boldsymbol{W}) = J(y^K(\boldsymbol{W}; x), \nabla_x y^K(\boldsymbol{W}; x))$, where $J$ is a differentiable loss function and $\|x\|_2 \leq C$. We assume that the activation function $h : \mathbb{R} \to \mathbb{R}$ has bounded first and second derivatives, i.e. $h(0) = 0, |h'(z)| \leq L_h, \quad |h''(z)| \leq M_h, \quad \forall z \in \mathbb{R}$. The loss function $J : \mathbb{R} \times \mathbb{R}^d \to \mathbb{R}$ has bounded second derivatives, i.e., $\|\nabla J(u, v)\| \leq M_J, \quad \forall u \in \mathbb{R}, v \in \mathbb{R}^d$, where $\nabla J(u, v)$ denotes the gradient of $J$ with respect to $(u, v)$. We assume $\max\{\|\boldsymbol{W}_1\|_{op, 2 \to RMS}, \sum_{i=2}^{K-1} \|\boldsymbol{W}_i\|_{op, RMS \to RMS}, \|\boldsymbol{W}_K\|_{op, RMS \to \infty}\} \leq C$. Then, the gradient of objective function $f(\boldsymbol{W}_{1:K})$ can be bounded in nuclear norm, i.e.,*

$$\|\nabla_{W_{1:K}} f(\boldsymbol{W}_{1:K})\|_{block, *} \leq O_{L_h, M_h, M_J, L}(\sqrt{m}), \tag{8}$$

*where $\| \cdot \|_{block, *}$ is the dual norm of the Block norm.*

**Why MUON helps and Why PINN is hard?** Unlike updates based on Euclidean geometry, such as the Frobenius norm, which scale with the sum of all $m^2$ gradient elements and can therefore grow as $O(m^2)$, MUON updates avoid this issue and improve that to $O(\sqrt{m})$. Under the Frobenius norm, larger networks require much smaller learning rates to maintain stable training. At the same time, as shown in Remark 2, if the objective function $J$ does not depend on the network's derivatives, as is the case in standard neural networks rather than PINNs, *i.e.* $f(\boldsymbol{W}_{1:K}) = J(y^K(\boldsymbol{W}_{1:K}; x))$, the nuclear norm of the gradient of $\nabla_{\boldsymbol{W}_{1:K}} f(X)$ is independent of the neural network width. This results also provides another explanation of why optimizing PINN is much harder than optimizing standard neural networks, which makes uncovering a reliable scaling law both more challenging and more valuable.

## 6 CONCLUSION

This work establishes, for the first time, a scaling law for physics-informed neural networks (PINNs). By reframing optimization through non-Euclidean geometry and introducing scaled MUON, we resolved the severe ill-conditioning that has historically prevented PINNs from benefiting from larger architectures. Across canonical PDE benchmarks, we showed that training dynamics become scale-invariant, with loss decreasing predictably as width increases. This enabled models exceeding 1,000 neurons per layer, compared to the previous practical ceiling of 200–400, while achieving lower errors and more stable convergence.

On the theoretical side, we show that neural networks constrained by our rescaled operator norm remain universal approximators for Sobolev functions. Importantly, this norm constraint is independent of network width, whereas in the Euclidean (Frobenius) geometry, the approximation norm grows with width. This demonstrates that the rescaled operator norm offers a more meaningful characterization of neural networks for Sobolev function approximation. We also theoretically show that MUON achieves this by carefully choosing appropriate norms for each layer, effectively controlling the size of parameter updates. In particular, the magnitude of updates in MUON corresponds to the nuclear norm of the gradient, which can be effectively bounded for PINNs.

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

# A  PROOF OF THE THEOREMS

## A.1  NORM CONSTRAINED APPROXIMATION

### A.1.1  PRELIMINARY ON APPROXIMATION THEORY

Our proof of the approximation upper bound is based on the observation that the B-spline approximation [12, 64] can be formulated as a ReLU3 neural network efficiently[70, 25, 14, 34]. Although the proof of the approximation of the neural network to the Sobolev spaces is a standard approach, we still demonstrate the proof sketch here.

**Definition 1.** *(Univariate and Multivariate B-splines) Fix an arbitrary integer $l \in \mathbb{Z}^+$. Consider a corresponding uniform partition $\pi_l$ of $[0,1]$:*

$$\pi_l : 0 = t_0^{(l)} < t_1^{(l)} < \cdots < t_{l-1}^{(l)} < t_l^{(l)} = 1,$$

*where $t_i^{(l)} = \frac{i}{l}$ ($\forall\, 0 \le i \le l$). Now for any $k \in \mathbb{Z}^+$, we can define an extended partition $\pi_{l,k}$ as:*

$$\pi_{l,k} : t_{-k+1}^{(l)} = \cdots t_{-1}^{(l)} = 0 = t_0^{(l)} < t_1^{(l)} < \cdots < t_{l-1}^{(l)} < t_l^{(l)} = 1 = t_{l+1}^{(l)} = \cdots = t_{l+k-1}^{(l)}$$

*Based on the extended partition $\pi_{l,k}$, the univariate B-splines of order $k$ with respect to partition $\pi_l$ are defined by:*

$$N_{l,i}^{(k)}(x) := (-1)^k (t_{i+k}^{(l)} - t_i^{(l)}) \cdot \left[ t_i^{(l)}, \cdots, t_{i+k}^{(l)} \right] \max\{(x-t), 0\}^{k-1}, \ x \in [0,1], \ i \in I_{l,k} \quad (9)$$

*where $I_{l,k} = \{-k+1, -k+2, \cdots, l-1\}$ and $\left[ t_i^{(l)}, \cdots, t_{i+k}^{(l)} \right]$ denotes the divided difference operator.*

*Equivalently, for any $x \in [0,1]$, we can rewrite the univariate B-splines $N_{l,i}^{(k)}(x)$ in an explicit form:*

$$N_{l,i}^{(k)}(x) = \begin{cases} \frac{l^{k-1}}{(k-1)!} \sum_{j=0}^{k} (-1)^j \binom{k}{j} \max\left\{ x - \frac{i+j}{l}, 0 \right\}^{k-1}, \ (0 \le i \le l-k+1) \\ \sum_{j=0}^{k-1} a_{ij} \max\left\{ x - \frac{j}{l}, 0 \right\}^{k-1} + \sum_{n=1}^{k-2} b_{in} x^n + b_{i0}, \ (-k+1 \le i \le 0) \\ \sum_{j=l-k+1}^{l} c_{ij} \max\left\{ x - \frac{j}{l}, 0 \right\}^{k-1}, \ (l-k+1 \le i \le l-1) \end{cases} \quad (10)$$

*where $\{a_{ij} \mid -k+1 \le i \le 0, \, 0 \le j \le k-1\}$, $\{b_{in} \mid -k+1 \le i \le 0, \, 1 \le n \le k-2\}$ and $\{c_{ij} \mid l-k+1 \le i \le l-1, \, l-k+1 \le j \le l-1\}$ are some fixed constants.*

*For any index vector $\boldsymbol{i} = (i_1, i_2, \cdots, i_d) \in I_{l,k}^d$, we can define a corresponding multivariate B-spline as a product of univariate B-splines:*

$$N_{l,\boldsymbol{i}}^{(k)}(\boldsymbol{x}) := \Pi_{j=1}^{d} N_{l,i_j}^{(k)}(x_j). \quad (11)$$

**Definition 2.** *(Interpolation Operator Schumaker (2007)) Take some domain $\Omega \subset [0,1]^d$ and two arbitrary integers $k, l \in \mathbb{Z}^+$. Consider the extended partition $\pi_{l,k}$ and the corresponding set of multivariate B-splines $\{N_{l,\boldsymbol{i}}^{(k)}(x)\}_{\boldsymbol{i} \in I_{l,k}^d}$ defined in Definition 1. For any $\boldsymbol{i} \in I_{l,k}^d$, we define the domain $\Omega_{\boldsymbol{i}} := \{\boldsymbol{x} \in \Omega : x_j \in [t_{i_j}, t_{i_j+k}], 1 \le j \le d\}$. There exists a set of linear functionals $\{\lambda_{\boldsymbol{i}}\}_{\boldsymbol{i} \in I_{k,l}^d}$, where $\lambda_{\boldsymbol{i}} : L^1(\Omega) \to \mathbb{R}$ ($\forall\, \boldsymbol{i} \in I_{k,l}^d$), such that for any $\boldsymbol{i} \in I_{k,l}^d$ and $p \in [1, \infty]$, we have:*

$$\lambda_{\boldsymbol{i}}(N_{l,\boldsymbol{j}}^{(k)}) = \delta_{\boldsymbol{i},\boldsymbol{j}} \text{ and } |\lambda_{\boldsymbol{i}}(f)| \le 9^{d(k-1)} (2k+1)^d \left(\frac{k}{l}\right)^{-\frac{d}{p}} \|f\|_{L^p(\Omega_{\boldsymbol{i}})}, \ \forall\, f \in L^p(\Omega). \quad (12)$$

*The corresponding interpolation operator $Q_{k,l}$ is defined as:*

$$Q_{k,l} f := \sum_{\boldsymbol{i} \in I_{k,l}^d} \lambda_{\boldsymbol{i}}(f) N_{l,\boldsymbol{i}}^{(k)}, \ \forall\, f \in L^1(\Omega).$$

**Theorem 3.** *[Schumaker (2007)] Fix $f \in W^s(\Omega)$ with $\Omega \subseteq [0,1]^d, s \in \mathbb{Z}^+$ and $p \in [1, \infty)$. Then for any $k, l, r \in \mathbb{Z}^+$ with $k \ge s$ and $0 \le r \le s$, we have that there exists some constant $C = C(k, s, r, p, d)$, such that:*

$$\|f - Q_{k,l} f\|_{H^r(\Omega)} \le C \left(\frac{1}{l}\right)^{s-r} \|f\|_{H^s(\Omega)}.$$

### A.1.2 Norm Based Universal Approximation

**Theorem 4.** *(Norm Constrained Approximation result of Deep Neural Network) Fix dimension $d \in \mathbb{Z}^+$ and domain $\Omega \subseteq [0,1]^d$, for any $s \in \mathbb{Z}^+$ and any function $u^* \in H^s(\Omega)$, there exists a spectral norm constrained Deep Neural Network $u_{DNN} \in \mathcal{F}_{ReLU^3}(K, m, B)$ with $K = O(1), m = O(N), B = O(1)$, such that:*

$$\|u_{DNN} - u^*\|_{L^2(\Omega)} \lesssim N^{-\frac{s}{d}} \|u^*\|_{H^s(\Omega)}. \tag{13}$$

*Proof.* Pick some $l = N^{\frac{1}{d}} \geq 2$. We firstly show that the given function $u^*$ can be approximated well by some linear combination of multivariate splines, which is denoted by $u_{\rm sp}$. Note that $N$ is assumed to be sufficiently large. Hence, we may pick $l = \lceil N^{\frac{1}{d}} \rceil = \Theta(N^{\frac{1}{d}}) \in \mathbb{Z}^+$ to be the partition size of the B-splines. Moreover, by picking $k = 4$ and $p = 2$ in Theorem 3, we have that the linear combination $u_{\rm sp} := Q_{4,l} u^* = \sum_{\boldsymbol{i} \in I_{4,l}^d} \lambda_i(u^*) N_{l,\boldsymbol{i}}^{(4)}$ satisfies:

$$\|u^* - u_{\rm sp}\|_{H^r(\Omega)} = \|u^* - Q_{4,l} u^*\|_{H^r(\Omega)} \leq C\Big(\frac{1}{l}\Big)^{s-r} \|u^*\|_{H^s(\Omega)} = C N^{-\frac{s-r}{d}} \|u^*\|_{H^s(\Omega)}.$$

We will then show that the linear combination $u_{\rm sp} = \sum_{\boldsymbol{i} \in I_{4,l}^d} \lambda_i(f) N_{l,\boldsymbol{i}}^{(4)}$ can be implemented by some Deep Neural Network $u_{\rm DNN} \in \mathcal{F}_{\rm ReLU^3}(K, m, B)$ with $K = O(1), m = O(N)$, and $B = O(1)$. Firstly, note that for $x \geq 0$, both $x$ and $x^2$ can be expressed in terms of the ReLU3 activation function $\eta_3$ with no error:

$$x = -\frac{1}{12}[\eta_3(x+3) - 5\eta_3(x+2) + 7\eta_3(x+1) - 3\eta_3(x) + 6]$$

$$x^2 = -\frac{1}{6}[\eta_3(x+2) - 4\eta_3(x+1) + 3\eta_3(x) - 4]$$

Applying the explicit formula listed in equation 10 implies that for any $-3 \leq i \leq l-1$, the univariate B-spline function $N_{l,i}^{(4)}(x)$ ($x \in [0,1]$) can be implemented by some ReLU3 Deep Neural Network $v_{\rm DNN}$ with both scalar input and scalar output. We have that for $v_{\rm DNN}$, the depth $L_v$ is 2 and the maximum width $W_v$ is upper bounded by 11.

Secondly, for any $x, y \geq 0$, we have that the product operation $x \cdot y$ can be expressed in terms of the ReLU3 activation function $\eta_3$ with no error:

$$x \cdot y = \frac{1}{2}[(x+y)^2 - x^2 - y^2]$$

$$= -\frac{1}{12}\Big[\eta_3(x+y+2) - 4\eta_3(x+y+1) + 3\eta_3(x+y)$$

$$- \eta_3(x+2) + 4\eta_3(x+1) - 3\eta_3(x) - \eta_3(y+2) + 4\eta_3(y+1) - 3\eta_3(y) + 4\Big]$$

In Schumaker (2007), it has been proved that the B-splines are always non-negative, i.e $N_{l,i}^{(4)}(x) \geq 0, \ \forall \ x \in [0,1]$. Therefore, by multiplying the non-negative univariate B-splines, we can implement any multivariate B-spline $N_{l,\boldsymbol{i}}^{(4)} = \Pi_{j=1}^d N_{l,i_j}^{(4)}(x_j)$ with some ReLU3 Deep Neural Network $p_{\rm DNN}$. We have that for $p_{\rm DNN}$, the depth $L_p = \lceil \log_2 d \rceil + 2$ and the maximum width $W_p = \max\{11d, \frac{9}{2}d\}$. Hence, we can further claim that $u^* = \sum_{\boldsymbol{i} \in I_{4,l}^d} \lambda_i(u^*) N_{l,\boldsymbol{i}}^{(4)}$, which is a linear combination of the multivariate B-splines $N_{l,\boldsymbol{i}}^{(4)}$, can be implemented by some ReLU3 Deep Neural Network $u_{\rm DNN}$. It remains to check that $u_{\rm DNN} \in \mathcal{F}_{\rm ReLU^3}(K, m, B)$ with $K = O(1), m = O(N)$ and $B = O(1)$. Note that we can ensure that the hidden layers of $u_{\rm DNN}$ are of the same dimension $W$ by adding inactive neurons.

For the depth $K$ of $u_{\rm DNN}$, we have that $K$ is equal to $L_p + 1$, where $L_p$ denotes the depth of the ReLU3 Deep Neural Network $p_{\rm DNN}$. Thus, we have $K = L_p + 1 = \lceil \log_2 d \rceil + 3$, which implies that $L = O(1)$. For the width $m$ of $u_{\rm DNN}$, we have that $m \leq |I_{k,l}^d| W_p$, where $W_p$ denotes the width of the ReLU3 Deep Neural Network $p_{\rm DNN}$. This implies:

$$m \leq |I_{k,l}^d| \times 11d = 11d(l+k)^d = 11d(l+4)^d = O(l^d) \Rightarrow m = O(N)$$

The norm constraint $B$ for $u_{\rm DNN}$ remains $O(1)$ because each neuron connects to only a fixed number of nodes in the next layer with constant weights. As a result, each neuron can be scaled by a constant factor, keeping the operator norm at $O(1)$. $\qquad\square$

## A.2 PROOF OF STABLE UPDATES

**Proof of Theorem 2.** In this section, we estimate the gradient scale of PINN under nuclear norm. To prove $\|\nabla_{W_{1:K}} f(W_{1:K})\|_{\text{block},*} \leq O_{L_h,M_h,M_J,L}(\sqrt{m})$, it is equivalent to prove the directional gradient $\nabla_{W_{1:K}} f(W_{1:K})[\Delta_{1:K}]$ is $O_{L_h,M_h,M_J,L}(\sqrt{m})$ for all $\|\Delta_{1:K}\|_{\text{block}} = 1$. For $\nabla_{W_{1:K}} f(W_{1:K})[\Delta_{1:K}] = \nabla_{y^K} J \cdot \nabla_{W_{1:K}} y^K(W_{1:K}; x)[\Delta_{1:K}] + \nabla_{\nabla_x y^K} J \cdot \nabla_{W_{1:K}} \nabla_x y^K(W_{1:K}; x)[\Delta_{1:K}]$, we only need to estimate $\nabla_{W_{1:K}} y^K(W_{1:K}; x)[\Delta_{1:K}]$ and $\nabla_{W_{1:K}} \nabla_x y^K(W_{1:K}; x)[\Delta_{1:K}]$.

*Proof.* Next, we begin by bounding the directional Hessian of the PINN. As a first step, we compute the full Hessian of the PINN, following the standard neural network computation procedure

$$y^i(W; x) = \begin{cases} h(y^{i-1}(W_{1:i-1}; x), W_i), & 2 \leq i \leq K, \\ h(x, W_1), & i = 1, \end{cases} \tag{14}$$

thus we have

$$\bullet \ \frac{\partial y^i}{\partial W_p} = \begin{cases} D_i \otimes (y^{i-1})^\top, & p = i, \\ D_i W_i \frac{\partial y^{i-1}}{\partial W_p}, & p < i. \end{cases}$$

Since $h(0) = 0$ and $|h'(z)| \leq L_h$, it follows that $|h(z)| \leq L_h|z|$. Moreover, given that $\|x\|_2 \leq C$ and $\max\left\{\|W_1\|_{\text{op}, 2\to\text{RMS}}, \ \sum_{i=2}^{K-1} \|W_i\|_{\text{op}, \text{RMS}\to\text{RMS}}, \ \|W_K\|_{\text{op}, \text{RMS}\to\infty}\right\} \leq C$, we obtain the bounds

$$\|y^i(W_{1:i}; x)\|_{\text{RMS}} \leq L_h^{i-1} C^i, \quad \forall \ 2 \leq i \leq K-1,$$

and $|y^K(W_{1:K}; x)| \leq L_h^{K-1} C^K$. Importantly, these upper bounds are independent of the network width $m$. Using the bound on $\|y^i(W_{1:i}; x)\|_{\text{RMS}}$, we can start to bound the directional gradient $\frac{\partial y^i}{\partial W_p}[\Delta]$ for $\|\Delta\|_{\text{op}} \leq 1$: (If $i = 1$, then the $\|\cdot\|_{\text{op}}$ should be changed to $\|\cdot\|_{\text{op},1\to\text{RMS}}$. All the proof preserves the same, thus we don't reply.)

- **Case 1** ($p = i$). Since $\frac{\partial y^i}{\partial W_i}[\Delta] = D_i(\Delta y^{i-1})$, thus we have $\|\partial_\Delta y\|_{\text{RMS}} \leq \|\operatorname{diag}(\sigma'(Wx))\|_{\text{op}} \|\Delta x\|_{\text{RMS}} \leq L_h \|y^i\|_{\text{RMS}} \leq L_h^i C^i$.

- **Case 2** ($p < i, i \neq K$). In this case, $y^i$ is a vector and we have:

$$\|\frac{\partial y^i}{\partial W_p}[\Delta]\|_{\text{RMS}} = \|D_i W_i \frac{\partial y^{i-1}}{\partial W_p}[\Delta]\|_{\text{RMS}} \leq L_h \|W_K\|_{\text{op}} \|\frac{\partial y^{i-1}}{\partial W_p}[\Delta]\|_{\text{RMS}}$$

$$\leq L_h C \|\frac{\partial y^{i-1}}{\partial W_p}[\Delta]\|_{\text{RMS}} \leq (L_h C)^{i-p} \|\frac{\partial y^p}{\partial W_p}[\Delta]\|_{\text{RMS}} \leq (L_h C)^i$$

- **Case 3** ($p < i = K$). In this case $y^i$ is a scalar and we have:

$$\left|\frac{\partial y^K}{\partial W_p}[\Delta]\right| = \left|D_i W_i \frac{\partial y^{K-1}}{\partial W_p}[\Delta]\right| \leq L_h \|W_K\|_{\text{op},\text{RMS}\to\infty} \|\frac{\partial y^{K-1}}{\partial W_p}[\Delta]\|_{\text{RMS}}$$

$$\leq L_h C \|\frac{\partial y^{K-1}}{\partial W_p}[\Delta]\|_{\text{RMS}} \leq (L_h C)^{K-p} \|\frac{\partial y^p}{\partial W_p}[\Delta]\|_{\text{RMS}} \leq (L_h C)^K$$

This means all $\frac{\partial y^i}{\partial W_p}[\Delta]$ can be bounded by $(L_h C)^i$ which is constant independent to the width $m$.

**Remark 2.** *This result also indicates that if the objective function $J$ does not depend on the network's derivatives, as is the case in standard neural networks rather than PINNs, i.e. $f(W_{1:K}) = J(y^K(W_{1:K}; x))$, the nuclear norm of the gradient of $\nabla_{W_{1:K}} f(X)$ is independent of the neural network width.*

We then bound the gradient corresponding to the derivative term in the PINN loss. Firstly we have

- For $\nabla_x y^1 = \boldsymbol{D}_1 W$, then we have

$$\|\nabla_{x_k} y^1\|_{\text{RMS}} = \|\boldsymbol{D}_1 W e_i\|_{\text{RMS}} \leq L_h \|W\|_{\ell_2 \to \text{RMS}} \|e_k\|_2 \leq L_h C.$$

- For $\nabla_x y^i(\boldsymbol{W}_{1:i}; x) = \text{diag}(\sigma'(\boldsymbol{W}_{1:i} y^{i-1}(\boldsymbol{W}; x))) \boldsymbol{W}_i \nabla_x y^{i-1}(\boldsymbol{W}_{1:i}; x)$, then we have

$$\|\nabla_{x_k} y^i(\boldsymbol{W}_{1:i}; x)\|_{\text{RMS}} = \|\boldsymbol{D}_i W_i \nabla_{x_k} y^{i-1}(\boldsymbol{W}_{1:i-1}; x)\|_{\text{RMS}} \leq L_h C \|\nabla_{x_k} y^{i-1}(\boldsymbol{W}_{1:i-1}; x)\|_{\text{RMS}}\| \leq (L_h C)^i.$$

This shows that the RMS norm of all $\nabla_x y^i$ can be bounded by a constant independent of the neural network width. Then we compute the gradient of the derivative term in the PINN loss. We have

- $\frac{\partial(\nabla_x y^i)}{\partial W_p}[\Delta] = \begin{cases} \text{diag}(\boldsymbol{D}_{i,2} \odot (\Delta \nabla_x y^{i-1})) \boldsymbol{W}_i \nabla_x y^{i-1} + \boldsymbol{D}_i \Delta \nabla_x y^{i-1}, & p = i, \\[2mm] \text{diag}\left(\boldsymbol{D}_{i,2} \odot \boldsymbol{W}_i \nabla_x y^{i-1}\right) \boldsymbol{W}_i \nabla_x y^{i-1} + \boldsymbol{D}_i W_i \frac{\partial(\nabla_x y^{i-1})}{\partial W_p}[\Delta], & p < i. \end{cases}$

Now we can start to bound the directional gradient $\frac{\partial \nabla_x y^i}{\partial \boldsymbol{W}_p}[\Delta]$ for $\|\Delta\|_{\text{op}} \leq 1$:

- **Case 1** $(p = i)$. Similarly, we can use $\|x \odot y\|_{\text{RMS}} \leq \sqrt{n}\|x\|_{\text{RMS}}\|y\|_{\text{RMS}}$ to bound the directional gradient as

$$\left\|\frac{\partial(\nabla_{x_k} y^i)}{\partial W_i}[\Delta]\right\|_{\text{RMS}} \leq \|\text{diag}(\boldsymbol{D}_{i,2} \odot (\Delta \nabla_{x_k} y^{i-1})) \boldsymbol{W}_i \nabla_{x_k} y^{i-1}\|_{\text{RMS}} + \|\boldsymbol{D}_i \Delta \nabla_{x_k} y^{i-1}\|_{\text{RMS}}$$
$$\leq L_h \|\nabla_{x_k} y^{i-1}\|_\infty \|W_i\|_{\text{op}} \|\nabla_{x_k} y^{i-1}\|_{\text{RMS}} + L_H \|\nabla_{x_k} y^{i-1}\|_{\text{RMS}}$$
$$\leq (L_h C \sqrt{m} \|\nabla_{x_k} y^{i-1}\|_{\text{RMS}} + L_h) \|\nabla_{x_k} y^{i-1}\|_{\text{RMS}} \lesssim (L_h C)^{2i-1} \sqrt{m}$$

- **Case 2** $(p < i, i \neq K)$. We can also do a similar computation which leads to the following bound:

$$\left\|\frac{\partial(\nabla_{x_k} y^i)}{\partial W_p}[\Delta]\right\|_{\text{RMS}}$$

$$\leq \|\text{diag}(\boldsymbol{D}_{i,2} \odot (\boldsymbol{W}_i \nabla_{x_k} y^{i-1})) \boldsymbol{W}_i \nabla_{x_k} y^{i-1}\|_{\text{RMS}} + \|\boldsymbol{D}_i \boldsymbol{W}_i \frac{\partial(\nabla_x y^{i-1})}{\partial W_p}[\Delta]\|_{\text{RMS}}$$

$$\leq L_h \|\boldsymbol{W}_i \nabla_{x_k} y^{i-1}\|_\infty \|W_i\|_{\text{op}} \|\nabla_{x_k} y^{i-1}\|_{\text{RMS}} + L_H C \|\frac{\partial(\nabla_x y^{i-1})}{\partial W_p}[\Delta]\|_{\text{RMS}}$$

$$\leq L_h \sqrt{m} \|\boldsymbol{W}_i \nabla_{x_k} y^{i-1}\|_{\text{RMS}} \|W_i\|_{\text{op}} \|\nabla_{x_k} y^{i-1}\|_{\text{RMS}} + L_H C \|\frac{\partial(\nabla_x y^{i-1})}{\partial W_p}[\Delta]\|_{\text{RMS}}$$

$$\lesssim L_h C^2 \sqrt{m} \|\nabla_{x_k} y^{i-1}\|_{\text{RMS}}^2 + L_H C \|\frac{\partial(\nabla_x y^{i-1})}{\partial W_p}[\Delta]\|_{\text{RMS}} \lesssim (L_h C)^{2i} \sqrt{m} + L H C \|\frac{\partial(\nabla_x y^{i-1})}{\partial W_p}[\Delta]\|_{\text{RMS}}.$$

- **Case 3** $(p < i, i = K)$. In this case $y^K$ is a scalar and all the computate is the same as **Case 2**. The reasoning follows

$$\left|\frac{\partial(\nabla_{x_k} y^i)}{\partial W_p}[\Delta] \nabla_{x_k} y^{i-1}\right|$$

$$\leq |\text{diag}(\boldsymbol{D}_{i,2} \odot (\boldsymbol{W}_i \nabla_{x_k} y^{i-1})) \boldsymbol{W}_i \nabla_{x_k} y^{i-1}| + \left|\boldsymbol{D}_i \boldsymbol{W}_i \frac{\partial(\nabla_x y^{i-1})}{\partial W_p}[\Delta]\right|$$

$$\leq L_h \|\boldsymbol{W}_i \nabla_{x_k} y^{i-1}\|_\infty \|W_i\|_{\text{op}, \text{RMS} \to \ell_\infty} \|\nabla_{x_k} y^{i-1}\|_{\text{RMS}} + L_H C \|\frac{\partial(\nabla_x y^{i-1})}{\partial W_p}[\Delta]\|_{\text{RMS}}$$

$$\lesssim L_h C^2 \|\nabla_{x_k} y^{i-1}\|_{\text{RMS}}^2 + L_H C \|\frac{\partial(\nabla_x y^{i-1})}{\partial W_p}[\Delta]\|_{\text{RMS}} \lesssim (L_h C)^{2i} + L H C \|\frac{\partial(\nabla_x y^{i-1})}{\partial W_p}[\Delta]\|_{\text{RMS}}.$$

Then we can have a bound $\left\| \frac{\partial(\nabla_{x_k} y^i)}{\partial W_p}[\Delta]\nabla_{x_k} y^{i-1} \right\|_{\text{RMS}} \leq i(L_h C)^{2i}\sqrt{m}$ which provides a $\sqrt{m}$ dependent bound

$\square$

## B  ADDITIONAL DETAILS ON SCALED MUON

**Derivation of last layer update.**  We only derive the update for the last layer in algorithm 1, which is induced by the $\|\cdot\|_{\text{op,RMS}\to\infty}$ norm. The derivation for the first layer follows similarly. Let $G \in \mathbb{R}^{m\times n}$ denote the matrix gradient. By definition, $\|G\|_{\text{op,RMS}\to\infty} = \sup_{\|d\|_{\text{RMS}}=1} \|Gd\|_\infty = \sqrt{n}\max_{1\leq i\leq m}\|G_{i,:}\|_2$, where $G_{i,:}$ denotes the $i$th row of $G$. Therefore, the corresponding unit-norm ball is the set of matrices $S$ whose rows are individually bounded in the Euclidean norm: $\|S_{i,:}\|_2 \leq \frac{1}{\sqrt{n}}$ for each row $i$. Since the steepest descent magnitude is attained at the boundary of the unit-norm ball, the steepest descent direction is obtained by normalizing each row of the gradient and scaling it by a factor: $-\frac{1}{\sqrt{n}}\frac{G_{i,:}}{\|G_{i,:}\|_2}$. In the last layer, $m$ equals the output dimension of the network and is constant as network width increases, so the magnitude of the weight update with respect to width is $\Theta(1/\sqrt{n})$, satisfying the spectral condition in (Yang et al., 2023).

**Runtime of experiments.**  The runtime of each training run is reported in table 3. Across multiple benchmark problems, we empirically show that Scaled MUON has similar runtime to SOAP.

Table 3: **Runtime of training runs (in hours).**  We compare the runtimes for each benchmark problem, optimizer, and network width. All optimizers were ran for 100,000 steps.

| | Widths | 8 | 16 | 32 | 64 | 128 | 256 | 512 | 1024 |
|---|---|---|---|---|---|---|---|---|---|
| | Scaled MUON | 0.34 | 0.38 | 0.34 | 0.36 | 0.48 | 0.44 | 1.36 | 6.23 |
| Burgers | Adam | 0.21 | 0.18 | 0.23 | 0.18 | 0.18 | 0.33 | 0.77 | 3.73 |
| | SOAP | 0.25 | 0.29 | 0.24 | 0.27 | 0.40 | 0.95 | 1.48 | 6.60 |
| | Scaled MUON | 0.29 | 0.31 | 0.33 | 0.32 | 0.33 | 0.44 | 1.40 | 6.33 |
| Allen-Cahn | Adam | 0.18 | 0.16 | 0.17 | 0.16 | 0.19 | 0.29 | 0.78 | 2.81 |
| | SOAP | 0.25 | 0.22 | 0.24 | 0.26 | 0.39 | 0.54 | 1.39 | 4.74 |
| | Scaled MUON | 1.08 | 1.17 | 0.98 | 1.14 | 0.95 | 1.31 | 3.19 | 12.09 |
| Kolmogorov Flow | Adam | 0.78 | 0.85 | 0.96 | 0.91 | 0.77 | 1.17 | 2.73 | 8.94 |
| | SOAP | 0.90 | 1.01 | 0.83 | 1.17 | 1.11 | 1.43 | 3.27 | 11.65 |
| | Scaled MUON | 0.42 | 0.39 | 0.41 | 0.48 | 0.42 | 0.54 | 1.64 | 7.39 |
| Wave | Adam | 0.28 | 0.28 | 0.30 | 0.24 | 0.29 | 0.42 | 1.13 | 3.89 |
| | SOAP | 0.35 | 0.33 | 0.48 | 0.40 | 0.50 | 0.75 | 1.76 | 5.75 |

**Algorithm 2** The Newton-Schulz algorithm for approximating the matrix sign function used in algorithm 1. Given a matrix $\tilde{G}$, the following function approximates $\text{signm}(\tilde{G})$ using an iterative method for $K$ steps.

> **function** NEWTON-SCHULZ($\tilde{G}, K$)
>     $X \leftarrow \tilde{G} \in \mathbb{R}^{m\times n}$
>     **if** $m > n$ **then**
>         $X \leftarrow X^\top \in \mathbb{R}^{n\times m}$
>     **for** $k = 1, ..., K$ **do**
>         $A \leftarrow XX^\top$         $\triangleright O(\min(m,n)^2\max(m,n))$
>         $B \leftarrow -1.5A + 0.5A^2$         $\triangleright O(\min(m,n)^3)$
>         $X \leftarrow 2X + BX$         $\triangleright O(\min(m,n)^2\max(m,n))$
>     **if** $m > n$ **then**
>         $X \leftarrow X^\top \in \mathbb{R}^{m\times n}$
>     **return** $X$

## C ADDITIONAL EXPERIMENT DETAILS

**Compute estimation.** In our experiments, we investigate how loss decreases with increasing network width in PINN training, focusing on the relationship between the loss and compute. Compute is taken to be of the same order as the total number of floating-point operations (FLOPs) required for training. Since the total compute equals the FLOPs per iteration times the number of training steps, it suffices to estimate the per-iteration cost. For MLP architectures, per-iteration cost is of the same order as a single forward pass, since backwards passes can be approximated as twice the cost of a forward pass. Therefore, we estimate compute by taking the number of trainable parameters times the number of training steps. In table 4, we show the number of trainable parameters for one of our experiments.

Table 4: Network parameters by width for a MLP in the 100D Poisson problem.

| Width | Trainable Parameters |
|---|---|
| 8 | 1,033 |
| 16 | 2,449 |
| 32 | 6,433 |
| 64 | 19,009 |
| 128 | 62,593 |
| 256 | 223,489 |
| 512 | 840,193 |
| 1024 | 3,253,249 |

**Scaling law parameter estimation.** To learn the parameters $L_0$, $A$, and $\alpha$, we solve the optimization problem

$$\min_{\ell_0, a, \alpha} \sum_{C_i} \text{Huber}_\delta(\log L_{\min}(C_i) - \text{LSE}(\ell_0, a - \alpha \log C_i)),$$

where $C_i$ are compute values from an interval and $L_0 = e^{\ell_0}$ and $A = e^a$. Here, LSE is the LogSumExp function defined as $\text{LSE}(x_1, ..., x_n) = \log(\exp(x_1) + \cdots + \exp(x_n))$, and $\text{Huber}_\delta$ is the Huber loss function defined as

$$\text{Huber}_\delta(a) = \begin{cases} \frac{1}{2}a^2 & \text{for } |a| \leq \delta, \\ \delta \cdot (|a| - \frac{1}{2}\delta) & \text{for } |a| > \delta \end{cases}$$

We set $\delta = 10^{-3}$ to mitigate the effect of outliers.

**Optimizer hyperparameters.** We report the key hyperparameters used for each optimization method in table 5. Since scaled MUON and MUON have the same hyperparameters, we omit reporting hyperparameters for MUON. The momentum parameter of scaled MUON and $\beta_1$ of Adam and SOAP are both first-order momentum terms, so we set them to the same values. Both scaled MUON and MUON use a Newton-Schulz iterative method to approximate the matrix sign function in the algorithm, and the number of Newton-Schulz steps determines how well the matrix sign function is approximated. For SOAP, the precondition frequency determines how often to update the preconditioners, so a lower precondition frequency allows for better optimization at the cost of extra computation.

**Loss weighting.** PINNs minimize a composite loss function of the form

$$\mathcal{L}(\theta) = \lambda_{\text{ic}}\mathcal{L}_{\text{ic}}(\theta) + \lambda_{\text{bc}}\mathcal{L}_{\text{bc}}(\theta) + \lambda_{\text{pde}}\mathcal{L}_{\text{pde}}(\theta),$$

where weights are used to balance the relative contributions to the overall loss. For Burgers, we set $\lambda_{\text{ic}} = 100$, $\lambda_{\text{bc}} = 100$, and $\lambda_{\text{pde}} = 1$. For Allen-Cahn, the loss function only contains the initial condition and PDE loss terms since we can enforce boundary conditions explicitly. We set $\lambda_{\text{ic}} = 100$ and $\lambda_{\text{pde}} = 1$. For Wave equation, there are two conditions at $t = 0$, one being the initial state and the other being the initial derivative. We set $\lambda_{\text{ic}} = 1000$, $\lambda_{\text{ic2}} = 1000$, $\lambda_{\text{bc}} = 1000$, and $\lambda_{\text{pde}} = 1$, where $\lambda_{\text{ic}}$ denotes the weight for the initial state and $\lambda_{\text{ic2}}$ denotes the weight for the

| Hyperparameter | Scaled MUON | Adam | SOAP |
|---|---|---|---|
| Learning rate | 1e-3 | 1e-3 | 1e-3 |
| Momentum | 0.95 | - | - |
| $(\beta_1, \beta_2)$ | - | (0.95, 0.999) | (0.95, 0.999) |
| Newton-Schulz steps | 30 | - | - |
| Precondition frequency | - | - | 1 |
| Weight decay | 0.0 | 0.0 | 0.0 |

Table 5: **Comparison of hyperparameters for Scaled MUON, Adam, and SOAP optimizers.**

initial derivative. For Kolmogorov flow, there are a total of five conditions. There are two initial conditions and two PDE conditions for $u$ and $v$, and there is one other constraint coming from the term $\nabla \cdot \mathbf{u} = 0$. We set all weights to 1.

**Scaled MUON on MLP architecture.** In figure 1, we show that larger PINNs are harder to train when using the PirateNet architecture. However, in figure 6, we achieve similar results but for MLPs, showing that PINN optimization instability exists across different architectures.

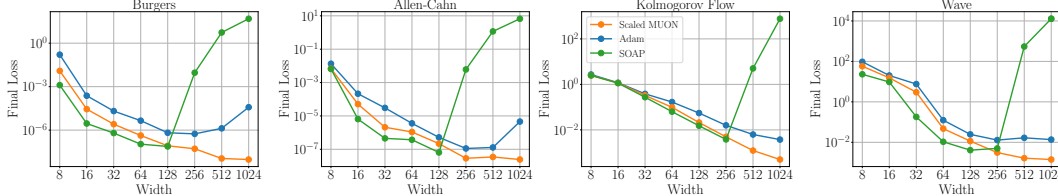

Figure 6: We replicate the experiment in Figure 1 with MLPs in place of PirateNets. Training instability at large widths persists, indicating that the phenomenon is not architecture-specific. However, Scaled MUON still enables scaling across different architectures.

## C.1 BENCHMARK PROBLEMS

Across benchmark problems, we provide additional figures that show the loss, boundary condition loss, initial condition loss, relative error. Additionally, we provide predictions and corresponding reference solutions for Scaled MUON at width 1024.

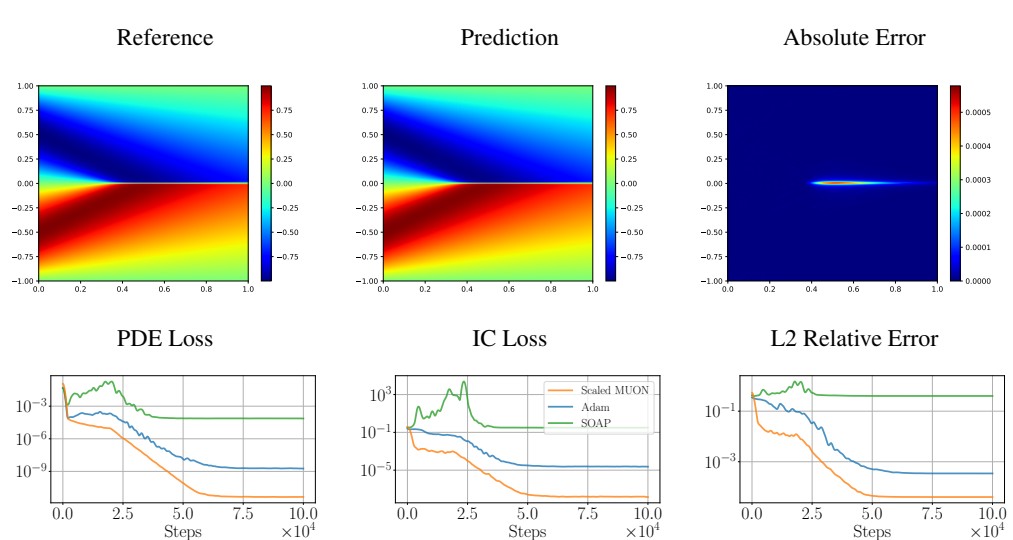

Figure 7: **Burgers.** *Top:* Comparison between reference solution and prediction using scaled MUON at width 1024. *Bottom:* Different training loss components and relative error between methods at width 1024.

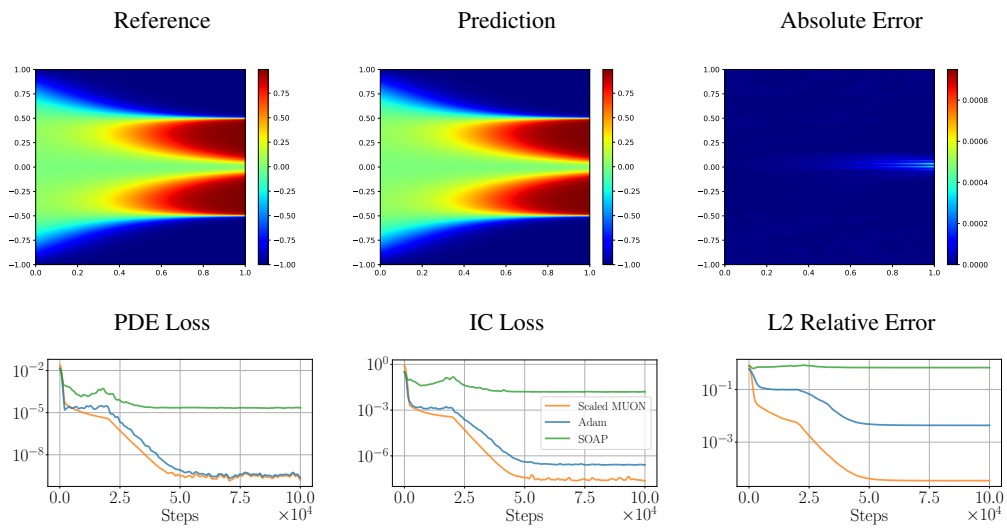

Figure 8: **Allen-Cahn.** *Top:* Comparison between reference solution and prediction using scaled MUON at width 1024. *Bottom:* Different training loss components and relative error between methods at width 1024.

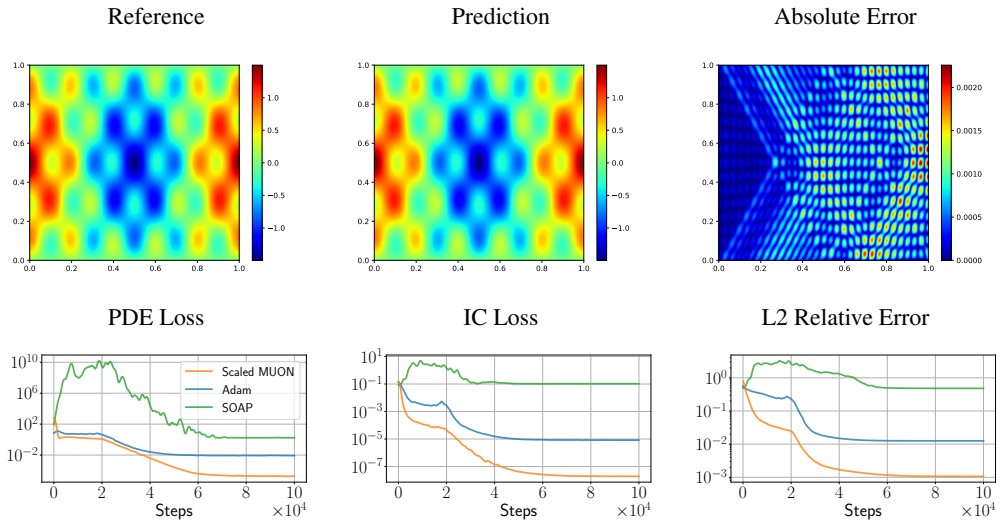

Figure 9: **Wave equation.** *Top:* Comparison between reference solution and prediction using scaled MUON at width 1024. *Bottom:* Different training loss components and relative error between methods at width 1024.

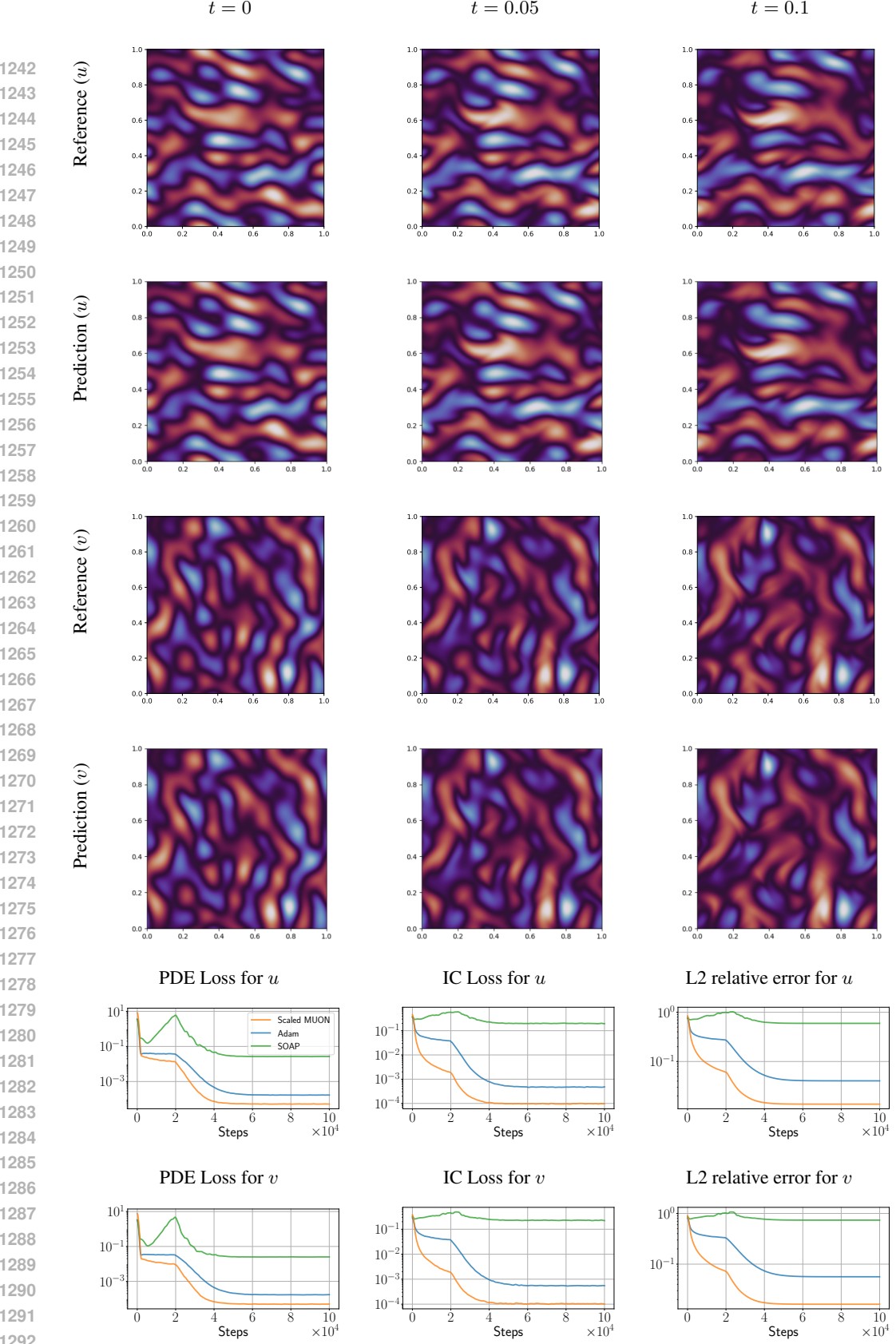

Figure 10: **Kolmogorov flow.** *Top:* Comparison between reference solution and prediction using scaled MUON at width 1024. *Bottom:* Different training loss components and relative error between methods at width 1024.

