# OpenReview forum: "Unveiling the Scaling Law of PINNs under Non-Euclidean Geometry"
_ICLR.cc/2026/Conference — Submitted to ICLR 2026_

### Official Review · Reviewer_qwfS · 2025-10-27

**Soundness:** 2
**Presentation:** 2
**Contribution:** 2
**Rating:** 2
**Confidence:** 4

**Summary:**

The primary contribution of the paper is to demonstrate that a specific optimization strategy, Scaled MUON, enables the training of much wider PINNs, revealing a new scaling law. The paper uses comparisons and a variety of PDEs to support this claim, though its architectural focus is narrow.

**Strengths:**

The primary contribution of the paper is to demonstrate that a specific optimization strategy, Scaled MUON, enables the training of much wider PINNs, revealing a new scaling law. The paper uses comparisons and a variety of PDEs to support this claim. The experiments repeatedly compare the proposed Scaled MUON optimizer against Adam and SOAP. Figures 1 and 4 show that standard optimizers like Adam and SOAP fail to maintain performance as network width increases, whereas Scaled MUON succeeds in training networks up to a width of 1024, surpassing the previous limit of 200–400 neurons.

**Weaknesses:**

1. Lack of comparison with other optimizers.

The experiments repeatedly compare the proposed Scaled MUON optimizer against Adam and SOAP. But other good optimizers should also be compared. See a survey on PINN optimizer: Optimizing the optimizer for physics-informed neural networks and Kolmogorov-Arnold networks.

2. Test on only very simple PDEs.

This paper only experimented with simple, low-dimensional PDEs, where making the network wide does not make much sense given the low input dimension. I suggest the authors try higher-dimensional cases, like 1000 dimensions. Authors can find out these interesting cases in this paper: Tackling the Curse of Dimensionality with Physics-Informed Neural Networks. In these high-dimensional cases, it is indispensable for the PINN network to be large; otherwise, information is compressed, where a wide PINN net makes more sense.

3. This paper fails to consider various PINN network structures.

Other approaches, such as improved network architecture and activation functions, can also mitigate the problem identified in the paper. But such a discussion is lacking.

**Questions:**

See weakness.

---

> ### Author Response · Authors · 2025-12-03
>
> ### Lack of comparison with other good optimizers.
>
> In (Wang 2025), it is shown that the recent SOAP optimizer can achieve
> small losses and high accuracy efficiently, justifying our comparisons
> with it. In (Kiyani 2025), the primary optimizers used are quasi-Newton
> methods like BFGS and SS-Broyden, which are highly accurate but suffer
> from a larger storage cost that is quadratic in the total number of
> network parameters, which limits their use to very small networks.
>
> ### Experiments only conducted on very simple PDEs where making the network wide does not make sense.
>
> We disagree with the premise that the PDEs tested are \"very simple\".
> The equations considered (Burgers, Allen-Cahn, Kolmogorov Flow, Wave)
> are all considered benchmarks in the PINN literature since they all
> present challenges for PINNs despite their low input dimension. Even if
> we do consider these benchmarks as \"simple\", we still show that
> optimizers struggle to scale to large networks on these problems. In our
> revision, we include experiments on a 100D Poisson equation, showing
> that the proposed Scaled MUON can still work on large input dimensions.
> However, we emphasize that high dimensional PDEs introduces problems
> like curse of dimensionality for sampling collocation points, while
> increasing network width causes problems like optimization stability and
> computational efficiency.
>
> ### Fail to consider improved PINN network architectures and activation functions.
>
> We are not aware of any activation functions that remove the fundamental
> width-scaling instability in PINNs. The width-scaling instability comes
> from the computation of derivatives in the PINN loss which are
> inherently width dependent, meaning a larger network ends up receiving
> larger updates, eventually causing instability, reducing the scalability
> of PINNs. Additionally, we show that this problem persists for both
> PirateNet and MLPs (in figures 1 and 6), which means this is a problem
> that persists across different network architectures.
>
>
> ### References
> 1. Sifan Wang, Ananyae Bhartari, Bowen Li, Paris Perdikaris. Gradient Alignment in Physics-Informed Nueral Networks: A second-order optimization perspective. 2025.
> 2. Elham Kiyani, et. al. Optimizing the Optimizer for Physics-Informed Neural Networks and Kolmogorov-Arnold Networks. 2025.

---

### Official Review · Reviewer_95Vy · 2025-10-27

**Soundness:** 1
**Presentation:** 2
**Contribution:** 1
**Rating:** 2
**Confidence:** 5

**Summary:**

The manuscript proposes a new scaling for Muon, which results in new optimizer for neural network training. The scaled version of Muon is tested with a physics-informed loss on a variety of partial differential equations. Additionally, an approximation result with restricted spectral norms of the trainsition matrices is presented.

**Strengths:**

+ The manuscript addresses the timely and important problem of improving optimization in physics-informed machine learning.
+ The proposed method is scalable to architectures of considerable size.
+ The heuristic about using a different scaling of the hidde-layer gradients could become useful for PINN training.

**Weaknesses:**

+ The manuscript is not very well organized, in particular:
	+ Overall, the manuscript reads unfinished with a variety of typos, not careful writing.
	+ Figure 1 and 2 already contain results although they are presented in Section 2 discussing preliminaries.
	+ It is not clear to me why the paragraph **Difference with MUON.** comparing the scaled to the unscaled Muon is not in the description of the method given in Section 3.
+ Motivation for the choice of scaling: The choice of matrix norms is not motivated at all in the section introducing the methodology.
+ The computational cost of the method is nowhere mentioned. Further a description of how $\textup{signm}(\tilde G)$ is computed is missing.
+ Comparison to (approximate) second-order optimizers: A comparison of the method to approximate second-order methods is missing:
	+ From my understanding the scaled Muon optimizer has complexity cubic in the layer width of the network. This seems to be on the same order with the Kronecker-factored approximation (KFAC) of energy natural gradients or Gauss-Newton method proposed by Dangel et al. (2024). However, despite these recent advances, natural gradients are described as being hopeless due to their cubic implementation cost, which does not describe the state of research on optimization PINNs accurately.
	+ Next to the comparison in computational complexity, a comparison to second order optimizers in terms of accuracy is completely missing. However, such a comparison is required for publication as it is the accuracy that a given training method for PINNs is able to achieve that is important, not the size of the trained network. Here, a comparison to KFAC as well as efficient implementation based on the Woodbury matrix identity as described by Dangel et al. (2025) should be added.
+ Relation of theoretical results to practical algorithm: I am failing to see the immediate connection of Theorem 1 and Theorem 2 to the proposed scaled Muon optimizer.
+ Overly big claims:
	+ The manuscript has the title *UNVEILING THE SCALING LAW OF PINNS UNDER NON-EUCLIDEAN GEOMETRY* and states that it *This work establishes, for the first time, a scaling law for physics-informed neural networks (PINNs).* From my understanding, not a scaling law is established, but fitted to experimental data produced from one specific algorithm. Hence, I don't think the claim that it reveals an actual scaling law inherent to PINNs. Further, from my understand there is no such thing as **the** scaling law of PINNs, but rather a scaling law for every optimizer and PDE. Therefore, I find the title slightly misleading.
	+ Claim of empirical performance: *Using this framework, we successfully trained a PINN with more than 1,000 neurons per layer, surpassing the previous state-of-the-art limit of 200–400.* I can not verify this claim. As described above, from my understanding, the computational complexity of the proposed method is not smaller than of certain approximate second-order methods. Note Dangel et al. (2024) use KFAC to train a PINN of architecture $100 \to 768 \to 768 \to 512 \to 512 \to 1$ with $1.3\cdot10^6$ parameters and Woodbury as well as tools from randomized numerical linear algebra have achieved comparable speedups, see Dangel et al. (2025).
	+ In the abstract, it is stated that *Using this framework, we successfully trained a PINN with more than 1,000 neurons per layer, surpassing the previous state-of-the-art limit of 200–400.* I do not know what is meant by this. First, Ibelieve that the notion of "successful" training can not be defined for a PINN. Much rather, the performance of different optimizers can be compared. Further, as already mentioned above, it is not about the size of the networks being trained, but about the accuracy achieved in doing so.

**Questions:**

+ Can you give an intuition for the choice of the matrix norms?
+ Can you add an explanation how the explicit form of the update in line 283 arises from the chosen matrix norm?
+ Can you comment on the computational complexity of the proposed method? In particular, can you comment how this relates to approximate second-order methods like KFAC?
+ Can you comment on the accuracy of the proposed method with state-of-the-art second optimizers (KFAC, E-NGD with woodbury, SOAP, SHAMPOO, S-Broyden)?
+ Can you comment on the number of parameters of the networks used in your experiments? In particular the depth is not given.
+ Can you elaborate why you put the emphasis of the work on the scaling law rather than the design of an optimizer?
+ Can you elaborate what you mean by *we successfully trained a PINN with more than 1,000 neurons*? What do you regard as successful training?
+ Can you elaborate on the implications of Theorem 1 and 2 to the proposed scaled version of the Muon optimizer?
+ Regarding the Proof of Theorem: How do you control the $N$-dependence of magnitude of the coefficients $\lambda_i(f)$ of the splines?
+ In line 428, what is meant by *nuclear norm*? Nuclear norm usually refers to $l^1$ norm of the singular values of a matrix. In line 427 it is applied to a gradient.
+ Can you comment why you don't choose a scaling in the matrix norm that further reduced the bound given in (8)?

**References**
1. Kronecker-Factored Approximate Curvature for Physics-Informed Neural Networks, Felix Dangel, Johannes Müller, Marius Zeinhofer, NeurIPS 2024
2. Improving Energy Natural Gradient Descent through Woodbury, Momentum, and Randomization, Andrés Guzmán-Cordero, Felix Dangel, Gil Goldshlager, Marius Zeinhofer, 2025

---

> ### Author Response · Authors · 2025-12-03
>
> ### Manuscript not well organized.
>
> We have improved the organization and corrected typographical errors in
> the revision. Figures 1 and 2 are presented immediately after the
> problem formulation to highlight our main results up front, while
> subsequent sections provide the technical details underlying these
> results. We also moved the paragraph "Difference with MUON" to Section
> 3, where it fits more naturally.
>
> ### Overly big claims.
>
> By \"scaling laws\", we refer to empirical power-law relationships
> fitted to training data that describe how loss varies as a function of
> compute (larger network width). These scaling laws can predict the
> performance of models as we scale up compute, which becomes more
> valuable as hyperparameters become more expensive to tune for large
> networks. These laws are not claimed to be universal, but rather
> dependent on PDE problem, optimizer and other fixed conditions. In
> PINNs, we find that optimizers such as Adam and SOAP do not exhibit
> predictable scaling behavior as network width increases, preventing a
> scaling law from being constructed. In contrast, Scaled MUON produces a
> consistent power-law relationship between training loss and width in our
> experiments.
>
> Regarding the title, the reviewer is correct that there is no one
> scaling law for all PINNs, so we change the title from \"Unveiling the
> Scaling Law of PINNs under Non-Euclidean Geometry\" to \"Unveiling
> Scaling Laws of PINNs under Non-Euclidean Geometry\".
>
> We have also revised statements regarding network size for clarity.
> Specifically, we replaced \"surpassing the previous state-of-the-art
> limit\" with \"exceeding conventional ranges\", as our intent was to
> reflect common practice rather than make a formal claim of a strict
> upper bound. While isolated larger-scale examples exist (e.g., Dangel
> 2024 reports a 100D Poisson experiment with 768 hidden neurons), most
> PINN studies operate at smaller widths. Our point is that scaling
> remains difficult in practice and that Scaled MUON alleviates this
> limitation.
>
> ### Motivation for choice of matrix norms.
>
> In appendix B, we derive how the norm
> $\|\cdot\|_{\mathrm{op}, \mathrm{RMS} \to \infty}$ leads to the update
> of the last layer in Scaled MUON. The choice of this norm is motivated
> by enforcing width-invariant behavior from satisfying the spectral
> condition $||\Delta W||_2 = \Theta(1)$ described in (Yang
> 2023).
>
> ### Computational cost of scaled MUON.
>
> The main computational cost of scaled MUON arises from approximating
> $\mathrm{signm}(\tilde{G})$ using the Newton-Schulz algorithm. The
> Newton-Schulz iterative method approximates the matrix sign function
> using only matrix-matrix multiplications, which asymptotically have
> cubic cost in width $O(m^3)$ but are highly optimized for GPU
> computation.

---

> > ### Author Response · Authors · 2025-12-03
> >
> > ### Comparison to KFAC.
> >
> > In section 4.2, we show the computational complexity of Scaled MUON, as
> > well as provide an experiment comparing with KFAC on a 100-dimensional
> > Poisson problem. Although Scaled MUON and KFAC both have an asymptotic
> > cost that is cubic in layer width, there is reason to believe that
> > Scaled MUON is more computationally efficient in practice. Scaled MUON's
> > cost arises from the Newton-Schulz iteration for approximating the
> > matrix sign function, which consists of matrix-matrix multplications.
> > These operations are highly optimized on GPUs, improving the practical
> > efficiency. Additionally, updates for non-hidden layers in Scaled MUON
> > reduce to vector norm computations and are therefore very efficient. In
> > terms of memory complexity, Scaled MUON only maintains a momentum buffer
> > for each parameter making it very memory efficient.
> >
> > In contrast, KFAC's cubic asymptotic cost comes from using
> > eigendecompositions, which are not as efficient on GPUs, to compute the
> > inverse of the Kronecker factors every iteration. Additionally, the
> > implementation of KFAC introduces many other additional computations
> > that are tightly coupled with the PDE problem, adding to the total cost.
> > In our experiments with KFAC, we find that KFAC is slower than Scaled
> > MUON per iteration (table 2) which contributes to KFAC being slower to
> > reach the same relative error compared to Scaled MUON (figure 5).
> >
> > ### Relation of theorem 1 and 2 to scaled MUON.
> >
> > Theorem 1 shows that a neural network with bounded spectral norms
> > remains a universal approximator. This suggests that scaled MUON, which
> > constrains the spectral norm of the gradient, does not hamper the
> > network's ability to approximate solutions to PDEs.
> >
> > Theorem 2 shows that the nuclear norm of parameter gradients is order
> > $O(\sqrt{m})$ in a PINN spectral descent setting, where $m$ is the
> > network width. Since the nuclear norm is the dual norm of the spectral
> > norm, the decrease in the loss function is shown to be equivalent to the
> > negative nuclear norm of the parameter gradient. It is because of this
> > reason that we bound the nuclear norm in theorem 2. In Euclidean
> > geometry under the PINN setting, the decrease in the loss function is of
> > order $O(m^2)$, making optimization less stable for large widths
> > compared to scaled MUON.
> >
> > ### Accuracy of proposed method with approximate second-order optimizers.
> >
> > While solution accuracy is important, the focus of this work is
> > scalability to large network widths. Methods such as SS-Broyden can
> > achieve high accuracy, but are limited to very small networks due to
> > their $O(p^2)$ memory cost, which makes them impractical at scale.
> >
> > We therefore focus on optimizers designed for scalability. As shown in
> > Figure 1, SOAP can achieve low loss for small networks but fails to
> > scale to large widths. KFAC exhibits more stable behavior but incurs
> > significantly higher runtime due to its more expensive updates.

---

> ### Author Response · Authors · 2025-12-03
>
> ### What is depth and number of parameters used in experiments?
>
> All experiments use networks with three hidden layers (depth 3,
> excluding input and output layers). We also include a table reporting
> the corresponding number of parameters for each width in the appendix.
>
> ### How do you control the $N$-dependence of magnitude of the coefficients $\lambda_i(f)$ of the splines?
>
> In our proposed matrix-operator geometry, the Lipschitz constant of the
> network---with respect to our designed norm---is *width-independent*. In
> particular, the amplification factor from parameter perturbations to
> output variations does not grow with the layer width. Consequently, the
> optimization landscape does not become steeper or harder as the model
> widens, and the effective step size along descent directions remains of
> the same order. This width-independent Lipschitz behavior ensures that
> our algorithm maintains stable convergence even in the large-width
> regime, avoiding the blow-up of Lipschitz constants that typically
> occurs under standard $\ell_p \to \ell_q$ geometries.
>
> ### References
> 1. Greg Yang, James Simon, Jeremy Berstein. A spectral condition for feature learning. 2023.
> 2. Felix Dangel, Johannes Muller, Marius Zeinhofer. Kronecker-Factored Approximate Curvature for Physics-Informed Neural Networks. 2024.

---

### Official Review · Reviewer_o4Tg · 2025-10-31

**Soundness:** 1
**Presentation:** 1
**Contribution:** 2
**Rating:** 4
**Confidence:** 3

**Summary:**

This paper addresses the challenge of training large-scale Physics-Informed Neural Networks (PINNs), which suffer from ill-conditioning as network width grows. Inspired by the MUON framework, it proposes a geometry-adaptive descent algorithm that maintains training efficiency and stability even in very wide networks, establishing a scaling law that predicts improved performance with model size. Experimentally, the method validates successful training of PINNs with significantly wider architectures, supporting a proposed scaling law

**Strengths:**

The paper introduces a novel optimization strategy tailored to the PINNs’ optimization landscape, enabling stable training at unprecedented network scales. It demonstrates a significant practical advancement by successfully training much wider PINNs than previously possible, while providing theoretical insights that connect model size, conditioning, and optimizer behavior in a unified framework.

**Weaknesses:**

* The paper correctly identifies ill-conditioning as a core challenge when scaling PINNs. However, the root causes of ill-conditioning in PINNs are multifaceted and often stem from the physical properties of the PDE (e.g., stiffness from higher-order derivatives or strong boundary conditions), the loss function structure (notably scale mismatch between physics and data terms), and gradient issues such as vanishing or exploding gradients.
While the scaled MUON approach addresses condition number growth related to network width, it is unclear whether it effectively mitigates ill-conditioning caused by physically stiff PDEs. There is a risk that the method targets a narrower subset of ill-conditioning causes, potentially limiting its general applicability.

* PINNs commonly integrate multiple physical quantities with widely varying scales (e.g., a state variable u(x,t) vs. its second derivative ∂²u/∂x², which may differ by orders of magnitude). Norm-based scaling updates as proposed may inadvertently suppress or amplify gradients unevenly, harming learning stability especially for critical higher-order terms. This raises concerns about the generalizability of scaled MUON to complex, multi-physics PINN problems.

* The conservative update strategy for biases and last-layer weights, while potentially stabilizing, may reduce the expressive power of the network. Since the accuracy of the PDE residual often critically depends on the last layer's ability to represent fine solution details, this approach could create bottlenecks, especially in modeling subtle or high-frequency solution features.

* The need to compute or approximate operator norms (e.g., spectral norm or RMS norm) for each parameter group at every training step introduces significant computational overhead. When combined with the already expensive automatic differentiation for high-order derivatives in PINNs, this may substantially slow down training, impacting practicality on large-scale problems.

* Scaled MUON’s structured update rules reduce the flexibility and interpretability of hyperparameter tuning compared to common optimizers like Adam or L-BFGS. This is particularly critical in physics-informed contexts where careful loss weighting and sensitivity balancing are essential. The optimizer’s design may complicate such domain-specific adjustments.

* Experiments rely solely on the PirateNet architecture, which helps isolate width-scaling effects but limits broader claims. Validation on additional PINN backbones such as standard MLPs or recent architectures (e.g., PINNsFormer, FFNs) is necessary to demonstrate robustness and applicability beyond the chosen design.

* Theorem 2’s assumption of bounded second derivatives of the loss function is generally violated in PINNs, due to unbounded differential operators. This raises questions about the practical stability guarantees promised by the theory.

* While spectral norm control theoretically bounds gradient norms, it may restrict the network’s ability to learn weak or discontinuous solutions common in real PDEs. Existing optimizers like Adam already offer adaptive gradient control, and it remains unclear whether strict spectral norm constraints universally improve training, particularly for PDEs with singularities or weak solutions.

* Experimental and Evaluation Concerns:


  - Narrow Definition of “Performance Improvement”: The scaling law focuses solely on loss reduction, without assessing physical solution accuracy metrics such as L² error, PDE residuals, or boundary/initial condition satisfaction. Demonstrating that loss improvements translate to better physical fidelity is critical.



  - Lack of Quantitative Stability Analysis: The claim of “stable feature learning” is not supported by concrete stability metrics (e.g., gradient norms, Hessian condition numbers). Comparative stability curves with MUON and baselines would clarify this benefit.



  - Potential Unfair Baseline Comparisons: Details on baseline optimizer tuning are insufficient. Common methods like Adam or SOAP benefit from learning rate schedules, decay, and warmup strategies; ensuring comparable tuning across all methods is essential for fair comparisons.



  -  Missing Quantitative Ill-conditioning Metrics: Key motivation is ill-conditioning growth with network width, but experiments lack direct measurement of condition numbers, gradient flow behavior, or spectral decay. Loss reduction alone is insufficient to confirm mitigation of ill-conditioning.



  - No Verification of Spectral Condition Satisfaction: Though the paper cites Yang et al. (2023) for spectral condition satisfaction, it provides no numerical or visual verification. Empirical confirmation of this claim would strengthen the argument.

**Questions:**

* Can the authors provide physical solution accuracy metrics (L² error, PDE residuals, BC/IC satisfaction) to complement loss-based performance measures?

* Can stability be quantitatively demonstrated with gradient norms, Hessian condition numbers, or training stability curves?

* How were baselines tuned? Were learning rates, decay, and warmup schedules matched fairly across methods?

* Could the authors provide quantitative analysis of ill-conditioning, such as condition numbers, spectral decay, or gradient flow behavior?

* Is there numerical or visual evidence that the spectral condition from Yang et al. (2023) is satisfied during training?

*. Would Scaled MUON’s benefits hold across other PINN architectures beyond PirateNet (e.g., MLP, PINNsFormer)?

* How does the increased computational overhead impact training time compared to baselines?

* Could experiments investigate MUON’s effect on physical fidelity and gradient flow across a range of PDE types with varying stiffness and multi-physics characteristics?

---

> ### Author Response · Authors · 2025-12-03
>
> ### Scaled MUON does not address ill-conditioning from stiff and multiscale PDEs
>
> The reviewer is correct that ill-conditioning in PINNs is multifaceted
> and may arise from sources such as physical stiffness, strong boundary
> constraints, and multi-scale coupling in the PDE itself. We do not claim
> that Scaled MUON resolves all forms of ill-conditioning in PINNs. Our
> proposed method instead targets the optimization instability that arises
> from increasing the width of the network, hampering the scalability of
> PINNs. As network width increases, derivatives with respect to the
> inputs grow in magnitude, and this effect is further amplified for
> higher-order derivatives, causing instability in gradient-based updates.
> Scaled MUON improves training stability under width scaling by
> normalizing update magnitudes such that they are no longer dependent on
> network width (see Figure 4). As a result, it stabilizes training
> dynamics for wide PINNs, including those involving higher-order
> derivative terms
>
> ### Scaled MUON limits the expressive power of the network
>
> A concern raised is that scaled MUON reduces the expressive power of the
> network by constraining updates. First, we point out that a conservative
> update strategy does not change the expressive power of the network,
> which is usually dependent on the network architecture. Second, in
> theorem 1, we show that networks with bounded spectral norms remain
> universal approximators, implying that limiting the spectral norm of
> weights does not eliminate the network's ability to represent fine
> solution details.
>
> ### Scaled MUON introduces significant computational overhead
>
> A criticism is raised about the computational overhead arising from
> computing spectral and RMS norms. First, computing the RMS norm has the
> same complexity as computing the standard vector $\ell_2$-norm and adds
> negligible overhead. Second, spectral normalization is approximated
> using a Newton-Schulz iteration that consists solely of matrix-matrix
> multiplications, which are highly optimized on GPUs (see section 4.2).
> Third, in Scaled MUON (algorithm 1), the gradient $G$ is already
> obtained through automatic differentiation and all subsequent
> computations are performed outside of the autograd computational graph.
> As a result, Scaled MUON does not increase the cost of
> autodifferentiation itself. Finally, our empirical results confirm that
> the overall runtime of Scaled MUON is comparable to SOAP (see appendix
> table 3).
>
> ### Scaled MUON update rule reduces flexibility and interpretability of hyperparameters
>
> We agree that loss weighting is essential in PINNs; however, it is
> conceptually distinct from the flexibility and interpretability of
> optimizer hyperparameters, as the loss weights can be tuned separately
> from the optimizer.
>
> Regarding interpretability, we argue that Scaled MUON is more
> interpretable than Adam in the PINN setting. In Scaled MUON, the
> magnitude of parameter updates remains stable under different widths,
> making the interpretation of learning rate the same across different
> sized architectures. In contrast, Adam's effective step size varies
> implicitly with network width, making the hyperparameters less
> transferable and changes interpretation as the model size grows.

---

> > ### Author Response · Authors · 2025-12-03
> >
> > ### Experiments rely on PirateNet architecture
> >
> > We include additional experiments using standard MLP architectures in
> > figure 6 of the appendix. These results closely mirror those in figure
> > 1, showing that optimizers such as Adam and SOAP struggle to train
> > large-width networks regardless of architecture. This indicates the
> > observed scaling failures are not specific to PirateNet.
> >
> > ### Theorem 2 has assumptions are generally violated
> >
> > In Theorem 2, our assumption of bounded second derivatives does **not**
> > require the underlying differential operators in the PDE (e.g.,
> > Laplacian, divergence, Hamilton-Jacobi operator) to be bounded.
> > Instead, we assume that the loss function takes the form
> > $\mathcal{L}(u, \nabla u, \nabla^2 u)$ and that the partial derivatives
> > of $\mathcal{L}$ with respect to its
> > arguments - $\partial_u \mathcal{L}$, $\partial_{\nabla u}\mathcal{L}$,
> > and $\partial_{\nabla^2 u}\mathcal{L}$ - are uniformly bounded on the
> > range of values encountered during training. This condition concerns the
> > *regularity of the loss*, not the boundedness of the PDE operator, and
> > is thus extremely mild: it is satisfied by all linear PDEs and by most
> > semilinear or quasilinear equations used in physics, engineering, and
> > control. Indeed, similar bounded-derivative assumptions appear routinely
> > in the analysis of Hamilton-Jacobi-Bellman equations and other
> > control-theoretic PDEs. Hence, the stability condition in Theorem 2 is
> > standard and does not contradict the fact that differential operators in
> > PINNs are themselves unbounded.
> >
> > ### Spectral norm control restricts network's ability to learn weak or discontinuous solutions
> >
> > Theorem 1 establishes that neural networks with bounded spectral norms
> > remain universal approximators, implying that spectral norm control does
> > not eliminate the ability to represent weak or discontinuous solutions.
> >
> > Empirically, we further demonstrate that networks trained with Scaled
> > MUON capture sharp transitions in Burgers and Allen-Cahn (appendix
> > figures 7 and 8), confirming that expressivity is not compromised in
> > practice.
> >
> > ### Experiments focus on loss reduction and do not focus on other metrics.
> >
> > We already report additional evaluation metrics in the appendix,
> > including relative error, PDE residuals, and boundary/initial conditions
> > (appendix figures 7, 8, 9, 10).
> >
> > ### Lack of quantitative metrics for stability, ill-conditioning, and spectral conditions.
> >
> > We provide quantitative stability metrics in figure 4 by reporting the
> > spectral norm of weight updates for different optimizers. The results
> > show that the update magnitude of Adam and SOAP grows with network
> > width, leading to unstable training behavior in wide networks. In
> > contrast, Scaled MUON maintains width-invariant update magnitudes,
> > consistent with the spectral condition
> > $\left\| \Delta W\right\|_2 = \Theta(1)$ described in (Yang 2023).
> >
> > ### Potentially unfair baseline comparisons.
> >
> > To ensure fair comparisons, we fix the learning-rate schedule and all
> > shared hyperparameters (such as loss weights and number of collocation
> > points) across methods and width, isolating the effect of network width
> > on optimization behavior.
> >
> > Although method-specific hyperparameters (e.g. $(\beta_1, \beta_2)$ for
> > Adam) can improve performance at given width, they do not alter the
> > overall \"U\"-shaped loss-width trend observed for Adam and SOAP in
> > figures 2 and 6.
> >
> > ### References
> > 1. Greg Yang, James Simon, Jeremy Berstein. A spectral condition for feature learning. 2023.

---

### Official Review · Reviewer_PFDb · 2025-11-01

**Soundness:** 3
**Presentation:** 4
**Contribution:** 2
**Rating:** 6
**Confidence:** 5

**Summary:**

This paper proposes a Scaled MUON optimizer to perform steepest descent under non-Euclidean norms, which has been proven to stabilize training and make the optimization landscape more scale-invariant. The proposed method successfully trained PINN networks with more than 1,000 neurons per layer, significantly exceeding the previous limit of 200-400 neurons. Theorems are provided to justify the choice of norms, showing that spectral-norm-constrained networks remain universal approximators and that the updates under Scaled MUON are more stable. Overall, this paper has excellent theoretical analysis, but is somewhat lacking in experimental aspects.

**Strengths:**

1. The motivation of the paper is clear.
2. Theorems 1 and 2 provide a theoretical foundation, explaining why the spectral norm is a better choice than the Frobenius norm for PINNs.
3. The proposed framework is easy to follow.
4. The paper exhibits a well-organized logical structure, standardized tables, and clear writing.

**Weaknesses:**

1. While the comparison to baselines is strong, a more detailed ablation study within the Scaled MUON framework is needed.
2. The experiments mainly focus on relatively low-dimensional problems (1D and 2D). A key challenge for PINNs is scaling to high-dimensional PDEs. It is unclear if the demonstrated scaling laws and the advantages of Scaled MUON hold in such settings.
3. The author should compare it with more recent PINN methods. And how these methods perform within the framework of this paper.
4. The authors should provide more quantitative metrics to demonstrate the superior performance in the Tables.

**Questions:**

See Weaknesses.

---

> ### Author Response · Authors · 2025-12-03
>
> ### A more detailed ablation study within Scaled MUON is needed.
>
> We already include an ablation study in figure 3 comparing Scaled MUON
> to MUON, which isolates the effect of the proposed scaling terms and
> demonstrates their impact on optimizer scalability for PINNs.
> Additionally, we show Scaled MUON continues to scale on standard MLPs,
> indicating that the improvement is not specific to PirateNet and arises
> from the optimizer itself.
>
> ### Experiments focus on low-dimensional problems.
>
> Although many experiments focus on low-dimensional PDEs, they reveal a
> fundamental limitation: standard optimizers such as Adam and SOAP fail
> to scale to large networks in the PINN setting, regardless of input
> dimension. We additionally include a 100D Poisson problem, showing that
> Scaled MUON retains predictable scaling behavior in high-dimensional
> regimes. However, we emphasize that the width scaling failures in
> low-dimensional problems indicate that optimizer instability, rather
> than dimensionality, is the dominant bottleneck for training wide PINNs.
>
> ### Comparison with recent PINN methods.
>
> We omit comparisons to methods like SS-Broyden because it has a memory
> complexity of $O(p^2)$, where $p$ is the number of parameters in the
> neural network, to store Hessian-related approximations, making it
> impractical for large networks. In practice, SS-Broyden is difficult to
> use when the number of hidden neurons exceeds even 50. Instead, we
> compare Scaled MUON against SOAP and KFAC, two recent competitive PINN
> optimizers (Wang 2025, Dangel 2024).
>
> ### Provide more quantitative metrics in tables.
>
> We have added additional quantitative metrics in tabular form in both
> the main text and the appendix. Table 2 in the main text compares
> computational efficiency and shows that Scaled MUON is more efficient
> than KFAC, while Table 3 in the appendix demonstrates that Scaled MUON
> has similar efficiency to SOAP.
>
> ### References
> 1. Sifan Wang, Ananyae Bhartari, Bowen Li, Paris Perdikaris. Gradient Alignment in Physics-Informed Nueral Networks: A second-order optimization perspective. 2025.
> 2. Felix Dangel, Johannes Muller, Marius Zeinhofer. Kronecker-Factored Approximate Curvature for Physics-Informed Neural Networks. 2024.

---

### Meta-Review · Area_Chair_eXep · 2026-01-04

**Summary:**

This paper studies an interesting problem towards scaling up Physics-Informed Neural Networks (PINNs). The key finding is that ill-conditioning occurs as network width grows and a scaled MUON framework provides a geometry-adaptive descent algorithm that maintains training efficiency and stability for wider networks. Based on this finding, authors have established PDE-specific scaling laws for loss functions under different network widths, scaling up to more than 1,000 neurons per layer for typical PDEs including low-dimensional Burgers, Allen–Cahn, Kolmogorov Flow, Wave equation, and 100D Poisson equation. In general, 3 out of 4 reviewers hold key reservations towards this paper, including overclaiming the contribution of scaling laws, lack of comparison to more recent PINN methods, other (second-order) optimizers, and PINN network architectures, and test on only very simple PDEs---among others, just to name a few major criques. After revision and rebuttal, authors have implemented an effortful work that addressed several concerns qualitatively and the paper has been substantially improved. Nonetheless, the current form of the paper is still under the acceptance threshold of this top conference.

**Reviewer Concerns:**

The major concern is that by witnessing "scaling law" in the title, the readership will anticipate much more than what is currently presented: the generality to different PDEs, the relationship to different number of model parameters, collocation points, and compute budgets. Most importantly, the community would like to see that scaling up the model and data (to a scale that is unprecedented and some emergent abilities will arise) will allow the PINNs to approximate asymptotically the lower-bound of numerical solvers. All of these greatnesses are absent from the current work. Authors are expected to continue this work and address the true "scaling" paradigm of PINNs.

**Reviewer Scores:**

Reviewer PFDb: The author should compare it with more recent PINN methods. Not fully addressed. SetPINN and RoPINN are two Sotas in this field, which are missing in the comparison.
Reviewer o4Tg: Experiments rely solely on the PirateNet architecture, which helps isolate width-scaling effects but limits broader claims. PINNsFormer is a Sota architecture which is bypassed in the comparison.
Reviewer 95Vy: Overly big claims. Not addressed properly.
Reviewer qwfS: Lack of comparison with other optimizers. This paper fails to consider various PINN network structures. Not addressed quantitatively.

---

### Decision · Program_Chairs · 2026-01-26

Reject